# Bayesian Analysis of Combinatorial Gaussian Process Bandits

**Jack Sandberg[1], Niklas Åkerblom[1,2] & Morteza Haghir Chehreghani[1]**
[1]Department of Computer Science and Engineering,
 Chalmers University of Technology & University of Gothenburg
[2]Volvo Car Corporation
`{jack.sandberg, morteza.chehreghani}@chalmers.se`,
`niklas.akerblom@volvocars.com`

## Abstract

We consider the combinatorial volatile Gaussian process (GP) semi-bandit problem. Each round, an agent is provided a set of available base arms and must select a subset of them to maximize the long-term cumulative reward. We study the Bayesian setting and provide novel Bayesian cumulative regret bounds for three GP-based algorithms: GP-UCB, GP-BayesUCB and GP-TS. Our bounds extend previous results for GP-UCB and GP-TS to the *infinite*, *volatile* and *combinatorial* setting, and to the best of our knowledge, we provide the first regret bound for GP-BayesUCB. Volatile arms encompass other widely considered bandit problems such as contextual bandits. Furthermore, we employ our framework to address the challenging real-world problem of online energy-efficient navigation, where we demonstrate its effectiveness compared to the alternatives.

## 1 Introduction

The multi-armed bandit (MAB) problem is a classical problem in which an agent repeatedly has to choose between a number of available actions to perform (commonly called *arms*) and receives rewards depending on the selected action. The goal of the agent is to minimize its expected cumulative regret over a certain time horizon, either finite or infinite, where regret is defined as the expected difference in reward between the agent's selected arm and the best arm (Robbins, 1985). The MAB problem has applications in healthcare, advertising, telecommunications and more (Bouneffouf et al., 2020).

The combinatorial MAB (Gai et al., 2012; Cesa-Bianchi & Lugosi, 2012; Chen et al., 2013) considers a problem where the agent must select a subset of the available base arms, a *super arm*, in every round. The large number of super arms necessitates efficient exploration and may require solving difficult optimization problems.

The arms and environments in bandit applications often have some side-information (or context) that is correlated with the reward, e.g., the titles or user specifications in a news recommendation problem. In the contextual MAB (Li et al., 2010; Krause & Ong, 2011; Agarwal et al., 2014; Zhou, 2016), before selecting an arm, the agent is provided a context vector (for the entire environment or each individual arm) that may (randomly) vary over time. By utilizing the information in the context the agent can learn the expected rewards more efficiently.

When the set of arms or contexts is continuous (infinite), it is necessary to impose smoothness assumptions on the expected reward since the agent can only explore a finite number of arms. A common assumption is that the expected reward is a sample from a Gaussian process (GP) over the arm or context set. For a sufficiently smooth GP, this ensures that arms which lie close in the arm space have similar expected rewards. Integrating GPs into bandits can yield higher sample efficiency and improved learning.

In Table 1, we provide an overview and comparison of similar work in GP MABs. The seminal paper of Srinivas et al. (2012) first introduced the GP-UCB algorithm, which combines *upper confidence bounds* (UCB) with GPs for MABs with finite or infinite arm sets. Srinivas et al. provided frequentist

Table 1: Comparison of similar work in GP MABs where $T$ is the horizon, $K$ is the maximum super arm size, and $\gamma_T$ ($\gamma_{TK}$) is the maximum information gain from $T$ ($TK$) base arms. Note that Takeno et al. (2023; 2024) obtain a regret bound of $\mathcal{O}(\sqrt{T\gamma_T})$ for IRGP-UCB, GP-TS and PIMS in the finite setting.

| | **Ours** | (Nika, 2022) | (Takeno, 2023; 2024) | (Kandasamy, 2018) | (Russo, 2014) | (Srinivas, 2012) |
|---|---|---|---|---|---|---|
| Infinite/Finite | Infinite | Infinite | Infinite | Infinite | Finite | Infinite |
| Volatile/Static | Volatile | Volatile | Static | Static | Volatile | Static |
| Combinatorial | ✓ | ✓ | ✗ | ✗ | ✗ | ✗ |
| Bayesian/Frequentist | Bayesian | Frequentist | Bayesian | Bayesian | Bayesian | Frequentist |
| GP-UCB | ✓ | ✗ | ✓ | ✗ | ✓ | ✓ |
| GP-TS | ✓ | ✗ | ✓ | ✓ | ✓ | ✗ |
| GP-BUCB | ✓ | ✗ | ✗ | ✗ | ✗ | ✗ |
| Regret | $K\sqrt{T\gamma_{TK}\log T}$ | $K\sqrt{T\gamma_{TK}\log T}$ | $\sqrt{T\gamma_T\log T}$ | $\sqrt{T\gamma_T\log T}$ | $\sqrt{T\gamma_T\log T}$ | $\sqrt{T\gamma_T\log T}$ |

regret bounds for GP-UCB on a MAB problem with a compact (infinite) arm space. Later work by Russo & Roy (2014) provided Bayesian regret bounds for GP-UCB and GP-TS, a similar algorithm based on Thompson sampling (Thompson, 1933), in the finite-arm setting with volatile arms. Volatile arms (often called *time-varying* or *sleeping* arms) means that not all arms are available to the agent in every round. This is a general formulation that encompasses other MAB extensions such as the contextual MAB.

For the infinite arm setting, Russo & Roy only hinted that the proof follows by discretization arguments. Using discretization, the recent work by Takeno et al. (2023; 2024) derives Bayesian regret bounds for GP-UCB and GP-TS in the infinite arm setting - but without volatile arms.

The combinatorial *and* contextual MAB with changing arm sets (C3-MAB) incorporates both extensions and has received much interest recently (Qin et al., 2014; Chen et al., 2018; Nika et al., 2020; 2022; Elahi et al., 2023), with applications in online advertisement, epidemic control and base station assignment (Nuara et al., 2018; Lin & Bouneffouf, 2022; Shi et al., 2023). Recent work by Nika et al. (2022) considered the C3-MAB with base arm rewards sampled from a GP. Nika et al. (2022) provided high probability regret bounds for a combinatorial variant of GP-UCB with an approximation oracle.

In this work, we present novel Bayesian regret bounds for both GP-UCB and GP-TS in the combinatorial volatile Gaussian process semi-bandit problem that extend previous regret bounds for GP-UCB and GP-TS to the infinite, volatile and combinatorial setting. As discussed above, our results hold for the contextual setting as it is a special case of the volatile setting. Additionally, we present a Bayesian regret bound for a third bandit algorithm called GP-BayesUCB (GP-BUCB) which is based on the BayesUCB algorithm of Kaufmann et al. (2012). Whilst GP-BUCB was introduced by Nuara et al. (2018) for a combinatorial bandit problem, to the best of our knowledge there are no regret bounds for GP-BUCB - even in the non-combinatorial setting. We demonstrate that the parametrization of GP-BUCB is more flexible than GP-UCB and is less prone to over-exploration whilst retaining theoretical guarantees.

Furthermore, we demonstrate the applicability of combinatorial and contextual GP bandits to large scale problems with experiments on an online energy-efficient navigation problem for electric vehicles on real-world road networks. With the increasing emergence of electric vehicles, addressing this problem is crucial to mitigating the so-called *range anxiety*. Åkerblom et al. (2023) introduced a combinatorial MAB framework using Bayesian inference to learn the energy consumption on each road segment. In this paper, we extend the framework of Åkerblom et al. to a contextual setting and apply it to real-world road networks. The experimental results demonstrate that the contextual GP model achieves lower regret than the Bayesian inference model.

Our contributions can be summarized as follows.

- We extend previous Bayesian regret bounds for GP-UCB and GP-TS to the *infinite*, *volatile* (previous results only held for finite volatile arms) and *combinatorial* setting.

- To the best of our knowledge, we establish the first regret bound for GP-BayesUCB.

- We develop a combinatorial and contextual bandit framework for the important real-world application of online energy-efficient navigation.

---

**Algorithm 1** Framework for combinatorial volatile semi-bandit problem

---

**Require:** Prior agent parameters $\boldsymbol{\theta}_0$, base arm set $\mathcal{A}$, super arm set $\mathcal{S}$, horizon $T$.
1: **for** $t \leftarrow 1, \ldots, T$ **do**
2:     $\mathcal{A}_t, \mathcal{S}_t \leftarrow \text{ObserveAvailableArms}(\mathcal{A}, \mathcal{S})$
3:     $\mathbf{U}_t \leftarrow \text{GetBaseArmIndices}(t, \boldsymbol{\theta}_{t-1})$
4:     $\mathbf{a}_t \leftarrow \text{SelectOptimalSuperArm}(\mathcal{S}_t, \mathbf{U}_t)$
5:     $\mathbf{r}_t \leftarrow \text{ObserveRewards}(\boldsymbol{a}_t)$
6:     $\boldsymbol{\theta}_t \leftarrow \text{UpdateParameters}(\boldsymbol{a}_t, \boldsymbol{r}_t, \boldsymbol{\theta}_{t-1})$

---

## 2 SETUP AND ALGORITHMS

In this section, we formulate our bandit problem, introduce Gaussian process bandit algorithms, and define the information gain, a quantity that is essential for GP bandits.

### 2.1 PROBLEM FORMULATION

We begin by formulating the combinatorial volatile Gaussian process semi-bandit problem. Let $\mathcal{A} \subset \mathbb{R}^d$ denote the set of base arms in a $d$-dimensional space and $\mathcal{S} = \{\mathbf{a} | \mathbf{a} \subset \mathcal{A}\} \subset 2^{\mathcal{A}}$ denote the set of feasible super arms. Note that $|\mathcal{A}|$ can either be finite or infinite, and $2^{\mathcal{A}}$ denotes the power set of $\mathcal{A}$. The expected reward for the base arms $f(a) \sim \mathcal{GP}(\mu(a), k(a, a'))$ is assumed to be a sample from a Gaussian process with mean function $\mu(a) : \mathcal{A} \to \mathbb{R}$ and covariance function $k(a, a') : \mathcal{A} \times \mathcal{A} \to [-1, 1]$.

At time $t$, a possibly random and finite[1] subset of base arms $\mathcal{A}_t \subseteq \mathcal{A}$ is available to the agent. In a combinatorial setting, the agent must select a feasible subset of base arms, a *super arm*, $\mathbf{a}_t \in \mathcal{S}_t$ where $\mathcal{S}_t \subset 2^{\mathcal{A}_t}$ is the set of feasible and available super arms. To facilitate a feasible combinatorial problem, the super arms have a maximum size $K$ ($|\mathbf{a}| \leq K \ \forall \mathbf{a} \in \mathcal{S}_t$). The agent observes the rewards of the selected base arms (semi-bandit feedback) $\mathbf{r}_t = \{r_{t,a} | a \in \mathbf{a}_t\}$ where the base arm reward $r_{t,a} = f(a) + \epsilon_{t,a}$ is a sum of the expected reward and i.i.d. Gaussian noise with zero mean and variance $\varsigma^2$. Motivated by the online energy-efficient navigation problem in Section 4.1, the total reward is assumed to be additive, and the agent also observes this reward at time $t$: $R_t = \sum_{a \in \mathbf{a}_t} r_{t,a}$. The total number of time steps, the horizon, is denoted by $T$. Let $H_t$ denote the history $(\mathcal{A}_1, \mathcal{S}_1, \mathbf{a}_1, \mathbf{r}_1, \ldots, \mathcal{A}_{t-1}, \mathcal{S}_{t-1}, \mathbf{a}_{t-1}, \mathbf{r}_{t-1}, \mathcal{A}_t, \mathcal{S}_t)$ of past observations and the currently available arms at time $t$.

In this work, we are interested in minimizing the Bayesian cumulative regret which, with a horizon of $T$, is defined as

$$\text{BR}(T) = \mathbb{E}\Big[ \sum_{t \in [T]} f(\mathbf{a}_t^*) - f(\mathbf{a}_t) \Big], \tag{1}$$

where $[T] := \{1, ..., T\}$, $\mathbf{a}_t^* = \arg\max_{\mathbf{a} \in \mathcal{S}_t} f(\mathbf{a})$ and $f(\mathbf{a}) = \sum_{a \in \mathbf{a}} f(a)$. As discussed by Russo & Roy (2014), allowing stochastic arm sets permits us to consider broader sets of bandit problems, an example of particular interest to us will be contextual models. Even though $\mathcal{A}_t$ is finite, note that the infinite case $|\mathcal{A}| = \infty$ is of great importance since it it is necessary for the context to be a continuous random variable.

Algorithm 1 provides a framework for the introduced bandit problem. In the framework, the agent parameters $\boldsymbol{\theta}_t$ are defined for a general agent and are not specified here. Similarly, $\mathbf{U}_t$ denotes the set of base arm indices which could be upper confidence bounds or a posterior sample, depending on the algorithm used.

### 2.2 BAYESIAN FRAMEWORK FOR COMBINATORIAL GAUSSIAN PROCESS BANDITS

A Gaussian process $f(a) \sim \mathcal{GP}(\mu(a), k(a, a'))$ is a collection of random variables such that for any subset $\{a_1, \ldots, a_N\} \subset \mathcal{A}$ the vector $\mathbf{f} = [f(a_1), \ldots, f(a_N)]$ has a multivariate Gaussian distribution.

---

[1]The restriction $|\mathcal{A}_t| < \infty$ prevents issues with limit points since the agent can only select the same base arm once. This limitation is not necessary in a non-combinatorial setting.

We take a Bayesian view of the combinatorial problem and consider $\mathcal{GP}(\mu, k)$ as a prior over the base arm rewards. GPs are very useful for defining and solving bandit problems, due to their probabilistic nature and the flexibility they provide through the design of suitable kernels.

Let $N_{t-1} = \sum_{\tau=[t-1]} |\mathbf{a}_\tau|$ denote the total number of base arms selected up to time $t-1$ and let $a_1, \ldots, a_{N_{t-1}}$ denote the arms selected before time $t$. Additionally, let $\mathbf{y} \in \mathbb{R}^{N_{t-1}}$ denote the corresponding observed base arm rewards and $\boldsymbol{\mu} = [\mu(a_1), \ldots, \mu(a_{N_{t-1}})]^\top$ denote the corresponding prior expected base arm mean rewards. Then, for any $a \in \mathcal{A}$, the posterior GP distribution is given by:

$$\mu_{t-1}(a) = \mu(a) + \mathbf{k}(a)^\top \left(\mathbf{K} + \varsigma^2 I\right)^{-1} (\mathbf{y} - \boldsymbol{\mu})^\top \tag{2}$$

$$k_{t-1}(a, a') = k(a, a') - \mathbf{k}(a)^\top \left(\mathbf{K} + \varsigma^2 I\right)^{-1} \mathbf{k}(a'), \tag{3}$$

where $\mathbf{K} = \left(k(a_i, a_j)\right)_{i,j=1}^{N_{t-1}}$ is the covariance matrix of the previously selected arms and $\mathbf{k}(a) = \left[k(a, a_1), \ldots, k(a, a_{N_{t-1}})\right]^\top$ is the covariance between $a$ and the previously selected arms. Let $\sigma_{t-1}(a)$ and $\sigma_{t-1}^2(a)$ denote the posterior standard deviation and variance respectively.

In 2012, Srinivas et al. introduced the GP-UCB algorithm, which selects the next arm based on an upper confidence bound. In our combinatorial setting, the GP-UCB algorithm selects the super arm $\mathbf{a}_t = \arg\max_{\mathbf{a} \in \mathcal{S}_t} U_t(\mathbf{a})$ where $U_t(\mathbf{a}) = \mu_{t-1}(\mathbf{a}) + \sqrt{\beta_t}\sigma_{t-1}(\mathbf{a})$, $\mu_{t-1}(\mathbf{a}) = \sum_{a \in \mathbf{a}} \mu_{t-1}(a)$, $\sigma_{t-1}(\mathbf{a}) = \sum_{a \in \mathbf{a}} \sigma_{t-1}(a)$ and $\beta_t$ is a confidence parameter, typically of order $\mathcal{O}(\log t)$. Kaufmann et al. (2012) introduced Bayes-UCB, which selects the arm with the largest $(1 - \eta_t)$-quantile, where the quantile parameter $\eta_t$ was of order $\mathcal{O}(1/t)$. Adapted to the combinatorial Gaussian process setting, we suggest the following selection rule for Bayes-UCB: $\mathbf{a}_t = \arg\max_{\mathbf{a} \in \mathcal{S}_t} \sum_{a \in \mathbf{a}} Q\left(1 - \eta_t, \mathcal{N}\left(\mu_{t-1}(a), \sigma_{t-1}^2(a)\right)\right)$, where $Q(p, \lambda)$ denotes the $p$-quantile of the distribution $\lambda$. We refer to this adapted version as GP-BUCB. Note that for $\lambda = \mathcal{N}(\mu, \sigma^2)$, the $p$-quantile is given by $Q(p, \mathcal{N}(\mu, \sigma^2)) = \mu + \sigma\sqrt{2}\,\text{erf}^{-1}(2p - 1)$ where $\text{erf}^{-1}(\cdot)$ is the inverse of the error function. Thus, GP-BUCB can be seen as a variant of GP-UCB where $\beta_t = 2\left(\text{erf}^{-1}(1 - 2\eta_t)\right)^2$. GP-TS (Russo & Roy, 2014; Chowdhury & Gopalan, 2017) selects the next arm randomly by using posterior sampling. If $\hat{f}_t(a) \sim \mathcal{GP}(\mu_{t-1}, k_{t-1})$ is a sample from the posterior distribution, then, in the combinatorial setting, GP-TS selects the super arm $\mathbf{a}_t = \arg\max_{\mathbf{a} \in \mathcal{S}_t} \hat{f}_t(\mathbf{a})$, where $\hat{f}_t(\mathbf{a}) = \sum_{a \in \mathbf{a}} \hat{f}_t(a)$.

## 2.3 Information Gain

The regret bounds of most GP bandit algorithms depend on a parameter called the maximal information gain $\gamma_T$ (Srinivas et al., 2012; Vakili et al., 2021). The maximal information gain (MIG) describes how the uncertainty of $f$ diminishes as the best set of sampling points $\mathbf{a} \subset \mathcal{A}, |\mathbf{a}| \leq T$ grows in size $T$. The MIG is defined using the mutual information between the observations $\mathbf{y_a}$ at locations $\mathbf{a}$ and expected reward function $f$:

$$\gamma_T := \sup_{\mathbf{a} \subset \mathcal{A}, |\mathbf{a}| \leq T} I(\mathbf{y_a}; f), \tag{4}$$

where $I(\mathbf{y_a}; f) = H(\mathbf{y_a}) - H(\mathbf{y_a}|f)$ and $H(\cdot)$ denotes the entropy. Both the true value of $\gamma_T$ and most upper bounds depend on the kernel function $k$ defining the GP from which $f$ is sampled from. Srinivas et al. (2012) initially introduced bounds on $\gamma_T$ for common kernels, such as the Matérn and RBF kernels. For the RBF kernel, Srinivas et al. showed that $\gamma_T = \mathcal{O}\left(\log^{d+1}(T)\right)$. Later, Vakili et al. (2021) presented a general method of bounding $\gamma_T$ that utilizes the eigendecay of the kernel $k$. Using this method, Vakili et al. obtained improved bounds on the Matérn kernel with smoothness parameter $\nu$: $\gamma_T = \mathcal{O}\left(T^{\frac{d}{2\nu+d}} \log^{\frac{2\nu}{2\nu+d}}(T)\right)$. To apply these bounds, we require that $\mathcal{A}$ is compact.

## 3 Regret Analysis

Whilst the work of Chen et al. (2013) can be seen as a standard combinatorial framework, we adopt the framework of (Russo & Roy, 2014) since it is better suited for Bayesian bandits with volatile and

infinite arms. Russo & Roy (2014) first provided a Bayesian regret bound for GP-UCB in a volatile (but non-combinatorial) setting with a finite arm set. Recently, Takeno et al. (2023) presented explicit proof for the Bayesian regret of GP-UCB with a compact and static arm set. In this section, we present novel Bayesian regret bounds for both GP-UCB and GP-TS in a combinatorial and volatile setting (including the contextual setting). Additionally, to the best of our knowledge, we present the first Bayesian regret bound for GP-BayesUCB. Similar to previous work, we first consider the finite arm case, $|\mathcal{A}| < \infty$, and then consider the infinite case, $|\mathcal{A}| = \infty$, by extending the finite arm results via a discretization.

## 3.1 FINITE CASE

We start by highlighting our technical contributions for GP-BUCB. Following the proof framework of Russo & Roy (2014), we seek to bound two terms: $\mathbb{E}[f(\mathbf{a}_t^*) - U_t(\mathbf{a}_t^*)]$ and $\mathbb{E}[U_t(\mathbf{a}_t) - f(\mathbf{a}_t)]$. For GP-BUCB, establishing an upper bound for the second term requires us to work around the non-elementary function $\mathrm{erf}^{-1}(u)$. Using Thm. 2 of Chang et al. (2011), we find that $\mathrm{erf}^{-1}(u) \geq \sqrt{-\omega^{-1}\log((1-u)/\vartheta)}$ for $\omega > 1$ and $0 < \vartheta \leq \sqrt{2e/\pi}\sqrt{\omega - 1}/\omega$, see Lemma A.13. The bound is tighter for larger values of the parameter $\vartheta$ (Chang et al., 2011), thus we set $\vartheta$ to its maximum value whilst $\omega$ is kept as a tunable parameter. Recall that the quantile parameter $\eta_t$ determines how quickly the confidence bound grows and the order $\xi > 0$ (s.t. $\eta_t = \mathcal{O}(t^{-\xi})$) is another tunable parameter. As shown in the lemma below, these parameters influence the bound we get.

**Lemma 3.1.** *Let* $C_\omega = \left(\sqrt{\pi}\omega/\sqrt{2e(\omega - 1)}\right)^{1/\omega}$, *then for GP-BUCB with confidence parameter* $\beta_t = 2\left(\mathrm{erf}^{-1}(1 - 2\eta_t)\right)^2$ *and* $\eta_t = \frac{\sqrt{2\pi}^\omega}{2|\mathcal{A}|^\omega t^\xi}$, $\xi > 0$, $\omega > 1$,

$$\sum_{t \in [T]} \mathbb{E}[f(\mathbf{a}_t^*) - U_t(\mathbf{a}_t^*)] \leq C_\omega \cdot \begin{cases} \frac{\omega}{\omega - \xi}T^{1 - \frac{\xi}{\omega}} & \text{if } \xi/\omega < 1, \\ 1 + \log T & \text{if } \xi/\omega = 1, \\ \frac{\xi}{\xi - \omega} & \text{if } \xi/\omega > 1. \end{cases}$$

Kaufmann et al. (2012) studied (non-GP and non-combinatorial) Bayes-UCB for a Bernoulli bandit with $\xi = 1$ whilst our analysis permit any $\xi > 0$. Lemma 3.1 shows that the ratio $\xi/\omega$ determines if the bound for the right term is sublinear, logarithmic or constant w.r.t $T$ for GP-BUCB. The equivalent bounds for GP-UCB and GP-TS are both constant if $\beta_t = 2\log\left(|\mathcal{A}|t^2/\sqrt{2\pi}\right)$, see Lemma A.2, thus we assume $\xi/\omega > 1$ to simplify the regret bounds.

Srinivas et al. (2012) showed, in a non-combinatorial setting, that the sum of posterior variances can be bounded by the information gain between the sampled points and the expected reward function $f$. Lemma 3 in Nika et al. (2022) (adopted to our setting in Lemma A.12) generalizes this result to a combinatorial setting. The result depends on the maximum eigenvalue of all possible posterior covariance matrices of size at most $K$, which we denote as $\lambda_K^*$.

Then, we present the main theorems for GP-UCB, GP-BUCB and GP-TS in the finite case, see Appendix A.1 for the proofs.

**Theorem 3.2** (Finite regret bounds). *Let* $C_K := 2(\lambda_K^* + \varsigma^2)$. *When* $\mathcal{A}$ *is finite, the Bayesian regret of*

(i) *GP-UCB with* $\beta_t = 2\log(|\mathcal{A}|t^2/\sqrt{2\pi})$ *is bounded as* $BR(T) \leq \frac{\pi^2}{6} + \sqrt{C_K TK\beta_T\gamma_{TK}}$.

(ii) *GP-BUCB with* $\beta_t = 2\left(\mathrm{erf}^{-1}(1 - 2\eta_t)\right)^2$ *for* $\eta_t = \frac{\sqrt{2\pi}^\omega}{2|\mathcal{A}|^\omega t^\xi}$, $\xi > \omega > 1$ *is bounded as* $BR(T) \leq \sqrt{C_K TK\beta_T\gamma_{TK}} + C_\omega \cdot \frac{\xi}{\xi - \omega}$ *where* $C_\omega = \left(\sqrt{\pi}\omega/\sqrt{2e(\omega - 1)}\right)^{1/\omega}$.

(iii) *GP-TS is bounded as* $BR(T) \leq \frac{\pi^2}{3} + 2\sqrt{C_K TK\beta_T\gamma_{TK}}$ *where* $\beta_t = 2\log\left(|\mathcal{A}|t^2/\sqrt{2\pi}\right)$.

For all three algorithms (if $\xi/\omega > 1$ for GP-BUCB), we find that $BR(T) = \mathcal{O}(\sqrt{\lambda_K^* TK\beta_T\gamma_{TK}})$ where $\gamma_{TK}$ is the MIG from $TK$ base arms. Using the bounds of $\gamma_T$ from Section 2.3, we get that the regret is sublinear in $T$ for both the RBF and Matérn kernels. The closest work, by Nika et al. (2022), obtains a frequentist regret bound of the same order. Nika et al. (2022) noted that $\lambda_K^* = \mathcal{O}(K)$ which gives a linear dependence on $K$ in the worst case (Zhan, 2005). For a linear kernel, the setting of

Wen et al. (2015) is similar to our setting and they obtain $\mathcal{O}(K\sqrt{\log K})$ and $\mathcal{O}(K)$ dependencies on $K$ whereas our dependency is $\mathcal{O}(K\sqrt{\log K})$. For combinatorial semi-bandits with linear reward functions (but independent arms), Merlis & Mannor (2020) obtain a $\Omega(\sqrt{K/\log K})$ lower bound which would suggest a gap of $\sqrt{K}\log K$ for the linear kernel. When $K = 1$, our results match the non-combinatorial results for GP-UCB. However, the improved random GP-UCB (IRGP-UCB) and GP-TS of Takeno et al. (2023; 2024) has a Bayesian regret of $\mathcal{O}(\sqrt{T\gamma_T})$ in the finite case, suggesting that a $\sqrt{\beta_T} = \mathcal{O}(\sqrt{\log T})$ improvement is possible. To our knowledge, there are no known lower bounds for the Bayesian regret of GP-bandit algorithms in general. However, for the SE-kernel the non-Bayesian regret satisfies $\Omega(\sqrt{T(\log T)^{d/2}})$ (Scarlett (2018); Cai & Scarlett (2021)). Taken at face value, this would imply that our bounds are tight up to logarithmic factors of $T$.

## 3.2 INFINITE CASE

The infinite case, $|\mathcal{A}| = \infty$, is an important generalization since many decision problems have a continuum of actions to select from. Based on our framing of contextual MABs as a subset of volatile MAB, an infinite arm set permits contexts with support on infinite domains such as continuous time. This setting is often analytically more difficult and requires the following additional assumptions:

**Assumption 3.3** (Regularity assumptions). *Assume $\mathcal{A} \subset [0, C_1]^d$ is a compact and convex set for some $C_1 > 0$. Furthermore, assume that $\mu$ and $k$ are both $L$-Lipschitz on $\mathcal{A}$ and $\mathcal{A} \times \mathcal{A}$, respectively, for some $L > 0$. In addition, for $f \sim \mathcal{GP}(\mu, k)$ assume that there exists constants $C_2, C_3 > 0$ such that:*

$$\mathbb{P}\left(\sup_{a \in \mathcal{A}} \left|\frac{\partial f}{\partial a^{(j)}}\right| > l\right) \leq C_2 \exp\left(-\frac{l^2}{C_3^2}\right), \tag{5}$$

*for $j \in \{1, \ldots, d\}$ and $l > 0$ where $a^{(j)}$ denotes the $j$-th element of $a$.*

Whilst the high probability bound on the derivatives of the sample paths is a common assumption in the literature (Srinivas et al., 2012; Kandasamy et al., 2018; Takeno et al., 2023), we additionally require that both $\mu$ and $k$ are Lipschitz but this is not particularly restrictive, see Remark A.6.

Following Srinivas et al. (2012), proofs for the compact case tend to use a discretization $\mathcal{D}_t \subset \mathcal{A}$ where each dimension is divided into $\tau_t$ points such that $|\mathcal{D}_t| = \tau_t^d$. Let $[a]_{\mathcal{D}_t}$ denote the nearest point in $\mathcal{D}_t$ for $a \in \mathcal{A}$ and similarly let $[\mathbf{a}]_{\mathcal{D}_t} = \{[a]_{\mathcal{D}_t} | a \in \mathbf{a}\}$ for $\mathbf{a} \subset \mathcal{A}$. Due to the assumption of volatile arms, we require the following finer discretization (as compared to Takeno et al., 2023):

**Assumption 3.4** (Discretization size). *Let $\tau_t$ denote the number of discretization points per dimension and assume that*

$$\begin{cases} \tau_t \geq 2t^2 KLdC_1(1 + tK\varsigma^{-1}), & \text{(6a)} \\ \tau_t/\beta_t \geq 8t^4 K^2 LdC_1, & \text{(6b)} \\ \tau_t^2/\beta_t \geq 8t^5 K^3 L^2 d^2 C_1^2 \varsigma^{-2}, & \text{(6c)} \\ \tau_t \geq t^2 KdC_1 C_3(\sqrt{\log(C_2 d)} + \sqrt{\pi}/2) & \text{(6d)} \end{cases}$$

*where the constants $C_1, C_2, C_3$ and $L$ are given by Assumption 3.3 whilst the constants $d, K$ and $\varsigma$ are defined by the bandit problem (Section 2.1).*

We note that Eq. (6d) is equivalent to the discretization size used by Takeno et al. (2023) with an extra factor of $K$ to account for the combinatorial setting whilst we introduce Eqs. (6a) to (6c) to bound $U_t([\mathbf{a}]_{\mathcal{D}_t}) - U_t(\mathbf{a})$. A key step to establish the regret bound of GP-UCB by Takeno et al. (2023) is to use the fact (for that setting) that $\mathbf{a}_t$ maximizes the upper confidence bound $U_t(\mathbf{a})$ and thus $U_t([\mathbf{a}_t^*]_{\mathcal{D}_t}) - U_t(\mathbf{a}_t) \leq 0$. Since we consider a setting with volatile arms, $[\mathbf{a}_t^*]_{\mathcal{D}_t}$ is not necessarily a feasible super arm and our technical contribution in the infinite setting is an analysis of the discretization error of $U_t([\mathbf{a}]_{\mathcal{D}_t}) - U_t(\mathbf{a})$.

**Lemma 3.5.** *If $U_t(\mathbf{a}) = \mu_{t-1}(\mathbf{a}) + \sqrt{\beta_t}\sigma_{t-1}(\mathbf{a})$, Assumption 3.3 holds and $\tau_t$ and $\beta_t$ satisfy Eqs. (6a) to (6c) in Assumption 3.4, then for any sequence of super arms $\mathbf{a}_t \in \mathcal{S}_t$ $t \geq 1$:*

$$\sum_{t \in [T]} \mathbb{E}\left[U_t([\mathbf{a}_t]_{\mathcal{D}_t}) - U_t(\mathbf{a}_t)\right] \leq \frac{\pi^2}{6}. \tag{7}$$

To bound the difference in posterior mean, $\mu_{t-1}([\mathbf{a}]_{\mathcal{D}_t}) - \mu_{t-1}(\mathbf{a})$, we Cholesky decompose $\mathbf{K} + \varsigma^2 I = \mathbf{L}\mathbf{L}^\top$ and note that $||\mathbf{L}^{-1}(\mathbf{y} - \boldsymbol{\mu})||_2$ has a chi distribution with at most $TK$ degrees of freedom. The difference in posterior standard deviation, $\sigma_{t-1}([\mathbf{a}]_{\mathcal{D}_t}) - \sigma_{t-1}(\mathbf{a})$, is bounded by using that $k$ is Lipschitz, Assumption 3.4 and other smaller steps.

Next, we present our regret bounds for GP-UCB, GP-BUCB and GP-TS in the infinite setting:

**Theorem 3.6** (Infinite regret bounds). *If Assumption 3.3 holds and $\tau_t$ satisfies Assumption 3.4, then the Bayesian regret of*

(i) *GP-UCB with $\beta_t = 2\log(\tau_t^d t^2/\sqrt{2\pi})$ is bounded as $BR(T) \leq \frac{\pi^2}{2} + \sqrt{C_K TK\beta_T \gamma_{TK}}$.*

(ii) *GP-BUCB with $\beta_t = 2\left(\mathrm{erf}^{-1}(1 - 2\eta_t)\right)^2$ for $\eta_t = (2\pi)^{\omega/2}/\left(2\tau_t^{d\omega} t^\xi\right)$, $\xi > \omega > 1$ is bounded as $BR(T) \leq \frac{\pi^2}{3} + \sqrt{C_K TK\beta_T \gamma_{TK}} + C_\omega \cdot \frac{\xi}{\xi - \omega}$ where $C_\omega = \left(\sqrt{\pi}\omega/\sqrt{2e(\omega - 1)}\right)^{1/\omega}$.*

(iii) *GP-TS is bounded as $BR(T) \leq \frac{2\pi^2}{3} + 2\sqrt{C_K TK\beta_T \gamma_{TK}}$.*

The proofs are presented in Appendix A.2. Similar to Takeno et al. (2023), the regret is decomposed into multiple terms which are either bounded by the finite case or by using results such as Lemma 3.5. Because of the stochastic arm selection, the regret for GP-TS must be decomposed into more terms compared to GP-UCB, which increases the constants in the bound. As in the finite case, we get that $BR(T) = \mathcal{O}(\sqrt{\lambda_K^* TK\beta_T \gamma_{TK}})$ for all three algorithms which matches the non-combinatorial result of Takeno et al. (2023) for $K = 1$.

## 4 EXPERIMENTS

In this section, we consider the important real-world application of online energy-efficient navigation for electric vehicles and formulate it as a combinatorial and contextual bandit problem. Previous work by Åkerblom et al. (2023) introduced a framework based on Bayesian inference to address the online navigation problem when no contextual information is available. Bayesian combinatorial bandits allow us to combine imperfect initial estimates with exploration to find efficient routes. In this work, we extend the framework to incorporate contextual information, enabling us to make use of correlations for even faster learning.

### 4.1 BANDIT FORMULATION OF ONLINE ENERGY EFFICIENT NAVIGATION PROBLEM

**The online energy-efficient navigation problem** Consider a directed graph $\mathcal{G}(\mathcal{V}, \mathcal{E})$ where the vertices $\mathcal{V}$ denote intersections of road segments and the edges $e = (u_1, u_2) \in \mathcal{E}$ denote the road segment from intersection $u_1$ to intersection $u_2$. Additionally, let $\mathcal{L}(\mathcal{G}) = \mathcal{G}(\mathcal{E}, \mathcal{C}_t)$ denote the directed line graph of $\mathcal{G}$ where the set of connections $\mathcal{C}_t \subseteq \{(e_1, e_2)|e_1 = (u, v) \in \mathcal{E}, e_2 = (v, w) \in \mathcal{E}\}$ determine which turns are legal in the road network at time $t$. Assume that we are given a start vertex $u_1 \in \mathcal{V}$ and a goal vertex $u_n \in \mathcal{V}$. Let $\mathcal{P}_t$ denote the set of simple feasible paths from $u_1$ to $u_n$ at time $t$. A path $\mathbf{p} = \langle u_1, u_2, \ldots, u_n \rangle$ is legal if all the connections are legal, and $\mathbf{p}$ is simple if every vertex is visited at most once. At each time step $t$, we observe the set of available paths $\mathcal{P}_t$ and a context vector $x_{t,e} \in \mathbb{R}^d$ for each edge $e \in \mathcal{E}$. The context $x_{t,e}$ can include static features, such as the length of the road segment, and time-varying features, such as the congestion level. Based on the available connections and the context vector, we select a path $\mathbf{p}_t \in \mathcal{P}_t$ and observe the energy consumption associated with each edge in the path (negated reward): $R_t = \sum_{e \in \mathbf{p}_t} r_{t,e}$. The goal of online energy-efficient navigation is to minimize the total energy consumed over a horizon $T$. Note that the base arm set $\mathcal{A}_t$ corresponds to all edge-context tuples $(e, x_{t,e})$ and that the base arm space is defined as $\mathcal{A} = \mathcal{E} \times \mathcal{X}$ where $\mathcal{X} \subseteq [0, C_1]^d$ is a compact and convex set for some $C_1 > 0$. The super arm set $\mathcal{S}_t$ corresponds to sequences of edge-context tuples that form paths in $\mathcal{P}_t$.

**Shortest paths with rectified Gaussians** Using regenerative braking, the energy consumption of an electric vehicle can be negative along individual road segments which presents challenges when we wish to find the most energy-efficient path. The most common shortest path algorithm, Dijkstra's algorithm (Dijkstra, 1959), does not permit negative edge weights. Whilst alternative shortest path

algorithms, such as Bellman-Ford (Shimbel, 1954; Bellman, 1958; Ford, 1956), allow negative edge weights, they are significantly slower and do not return a path if the graph has a reachable negative cycle. To avoid the complexity associated with negative weights, we use the rectified normal distribution to get non-negative energy consumption estimates $U_{t,e}$ as input for Dijkstra's algorithm.

Upper confidence bound methods output optimistic estimates $\tilde{\mu}_e \in \mathbb{R}$ whereas Thompson sampling outputs posterior estimates $\tilde{\mu}_e \in \mathbb{R}$ by sampling from the posterior $\mathcal{N}(\mu_{t-1,e}, \sigma^2_{t-1,e})$. To ensure non-negative weights, the edge weight $U_{t,e}$ is set to $\mathbb{E}[z_e]$ where $z_e$ is distributed as the rectified Gaussian $\mathcal{N}^R(\tilde{\mu}_e, \sigma^2_{t-1,e})$. A random variable $Y = \max(0, X)$ is said to have a rectified Gaussian distribution $\mathcal{N}^R(\mu, \sigma^2)$ if $X \sim \mathcal{N}(\mu, \sigma^2)$. In Algorithm 2, we show how to integrate UCB, BUCB and Thompson sampling with rectification within the framework of Algorithm 1. The notation $\mu_{t-1,e}$ and

---

**Algorithm 2** Compute Rectified Indices

**procedure** GETBASEARMINDICES($t, \mathcal{A}_t, \boldsymbol{\theta}_{t-1} = (\boldsymbol{\mu}_{t-1}, \boldsymbol{\sigma}_{t-1}, \boldsymbol{\varsigma}_{t-1})$)

1: **for** each edge $e \in \mathcal{A}_t$ **do**
2:     $\tilde{\mu}_e \leftarrow \mu_{t-1,e} - \sqrt{\beta_t}\sigma_{t-1,e}$             ▷ UCB
2:     $\tilde{\mu}_e \leftarrow Q(\frac{1}{t}, \mathcal{N}(\mu_{t-1,e}, \sigma^2_{t-1,e}))$     ▷ BUCB
2:     $\tilde{\mu}_e \leftarrow$ Sample from $\mathcal{N}(\mu_{t-1,e}, \sigma^2_{t-1,e})$    ▷ TS
3:     $U_{t,e} \leftarrow \mathbb{E}[z_e]$ where $z_e \sim \mathcal{N}^R(\tilde{\mu}_e, \varsigma^2_{t-1,e})$
4: **return** $\mathbf{U}_t$

---

$\sigma^2_{t-1,e}$ refer respectively to the posterior mean and variance of the expected energy consumption for edge $e$ whilst $\varsigma^2_{t-1,e}$ refers to the variance of the noise. Since the number of edges $|\mathcal{E}|$ may be large, each edge is sampled independently in TS, as by Nuara et al. (2018). Note that Algorithm 2 decouples the probabilistic regression model and Thompson sampling. In the next sections, we describe two probabilistic regression models for energy-efficient navigation.

## GP regression for energy-efficient navigation

To our knowledge, this study is the first combinatorial Gaussian process bandit solution for online energy-efficient navigation. The energy consumption depends on both the structure of the graph and the provided context. We use the graph Matérn kernel $k_G : \mathcal{E} \times \mathcal{E} \rightarrow \mathbb{R}$ from Borovitskiy et al. (2021) to encode the structure of the line graph $\mathcal{L}(\mathcal{G})$ into the GP and an ordinary 5/2-Matérn kernel $k_f : \mathcal{X} \times \mathcal{X} \rightarrow \mathbb{R}$ to encode the dependence on the context. The two kernels are combined $k_{G.f+f} = k_G \cdot k_f + k'_f$ where the two feature kernels $k_f$ and $k'_f$ use separate sets of lengthscale and outputscale parameters. The cubic cost of exact GPs prohibits their application to large datasets. The sparse variational Gaussian processes (SVGP) (Titsias, 2009; Hensman et al., 2013) approximate the

---

**Algorithm 3** SVGP Optimization Procedure

**procedure** UPDATEPARAMETERS($\mathbf{a}_t, \mathbf{r}_t, \boldsymbol{\theta}_{t-1}$)

1:    Add $\mathbf{a}_t, \mathbf{r}_t$ to history.
2:    Set inducing points $\mathbf{Z}_t$ to top $M$ most visited edges.
3:    **for** $i \in \{1, \dots, G\}$ **do**
4:      $\tilde{\mathbf{a}}, \tilde{\mathbf{r}} \leftarrow$ Subsample batch of size $B$ from history.
5:      Compute batch ELBO.
6:      Optimize variational parameters with NGD.
7:    Compute $\boldsymbol{\mu}_t, \boldsymbol{\sigma}_t, \boldsymbol{\varsigma}_t$ using the optimized GP.
8:    **return** $\boldsymbol{\mu}_t, \boldsymbol{\sigma}_t, \boldsymbol{\varsigma}_t$ and the optimized variational parameters.

---

posterior distribution using a set of inducing points $\mathbf{Z}_t = \{z_1, \dots, z_M\}$ where $z_i \in \mathcal{A}$ and $M$ is significantly smaller than the number of datapoints. By defining a prior distribution $q(\mathbf{u}_t) = \mathcal{N}(\mathbf{m}_t, \mathbf{S}_t)$ for the inducing variables $\mathbf{u}_t$, an approximate GP posterior can be obtained such that the complexity to perform $N$ predictions is $\mathcal{O}(M^2 N)$, i.e. linear w.r.t. $N$. The variational paramaters $(\mathbf{m}_t, \mathbf{S}_t)$ are optimized by minimizing the evidence lower bound (ELBO) by performing $G$ stochastic (natural) gradient descent steps using batch size $B$. Since the inducing points $z_i$ lie in a mixed discrete and continuous space ($\mathcal{A} = \mathcal{E} \times \mathcal{X}$ for $\mathcal{X} \subset [0, C_1]^d$), we heuristically set $z_i$ equal to the edge-context tuple of the $i$-th most visited edge at the start of the SVGP optimization. Then, the continuous dimensions of $z_i$ are optimized together with $(\mathbf{m}_t, \mathbf{S}_t)$ using natural gradient descent (NGD) (Salimbeni et al., 2018). The procedure is described in Algorithm 3. Further details of the kernels and parameter values are provided in Appendices B.1 and B.3.

**Bayesian inference for energy-efficient navigation**    Åkerblom et al. (2023) introduced a framework for energy-efficient navigation using Bayesian inference to learn the distribution of the energy consumption in each road segment. The key assumption is that the energy consumption of an electric vehicle driving along a road segment is stochastic and follows a Gaussian distribution with unknown

mean and known variance. Additionally, it is assumed that the energy consumption along different edges is independent. Using a Gaussian prior, the posterior distribution for edge $e$ is computed using standard conjugate update rules.

**Real-world road networks**  In our experiments we use the road networks of Luxembourg and Monaco (Codeca et al., 2017; Codeca & Härri, 2018, based on data by OpenStreetMap contributors, 2017). Elevation data (Administration de la navigation aérienne, 2018) is added to the network using QGIS and the *netconvert* tool from SUMO. In Fig. 1, the two road networks are visualized along with two evaluation routes (A and B) per network. The evaluation routes span multiple regions of the network, allowing for many alternative paths. The context for each road segment consists of three fixed scalar properties: the length, the speed limit and the incline.

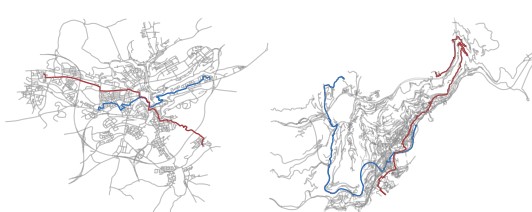

Figure 1: Road networks of Luxembourg (left) and Monaco (right) with evaluation routes A and B highlighted in blue and red.

Each property is standardized to have unit-variance. The prior expected energy consumption is computed by a deterministic model that assumes that the vehicle drives along an edge $e \in \mathcal{E}$, with length $\ell_e$ and inclination $\alpha_e$, at constant speed $v_e$. The expended energy is then

$$E_e^{\text{det}} := (mg\ell_e \sin(\alpha_e) + mgC_r\ell_e \cos(\alpha_e) + 0.5C_dA\rho\ell_e v_e^2)/3600\eta. \quad (8)$$

The deterministic energy consumption $E_e^{\text{det}}$ in Eq. (8) is given in Watt-hours and depends on the following vehicle-specific parameters: mass $m$, rolling resistance $C_r$, front surface area $A$, air drag coefficient $C_d$ and powertrain efficiency $\eta$. The gravitational acceleration $g$ and air density $\rho$ also determine $E_e^{\text{det}}$. The parameter values are specified in Table 2 in Appendix B.3. Let $\overline{E^{\text{det}}} = \frac{1}{|\mathcal{E}|}\sum_{e \in \mathcal{E}} E_e^{\text{det}}$ and $\sigma_{\text{det}}^2 = \frac{1}{|\mathcal{E}|}\sum_{e \in \mathcal{E}}(E_e^{\text{det}} - \overline{E^{\text{det}}})^2$ denote the mean and variance of the deterministic energy consumption. The expected energy consumption is sampled from $\mathcal{GP}(E^{\text{det}}, k_{G \cdot f + f})$ where the outputscale of $k_{G \cdot f + f}$ (i.e. the variance $\sigma_0^2$) is set to $0.25^2\sigma_{\text{det}}^2$. The noise variance $\varsigma^2$ is set to $0.1^2\sigma_{\text{det}}^2$ for all edges and the kernel lengthscales are set to 1. See Appendix B for further details.

## 4.2 RESULTS

Here, we demonstrate our experimental studies in different settings. We begin by comparing GP algorithms to Bayesian inference methods, then we compare the parametrizations of GP-UCB and GP-BUCB. Finally, we study the impact of the kernel lengthscale. Visualizations of the exploration are provided in Appendix C.2.

**Investigation of different bandit algorithms**  In our first experiment, we compare the three algorithms GP-UCB, GP-BUCB and GP-TS. We use the Bayesian inference (BI) method of Åkerblom et al. (2023) with UCB, BUCB and TS as baselines. For UCB and BUCB (GP and BI), we use the $\beta_t$ parametrization given by Theorem 3.2 with $\omega = 1, \xi = 1$. The six methods are evaluated 5 times each on the four routes in the Luxembourg and Monaco networks with a horizon of $T = 500$. The cumulative regret is shown in Fig. 2. The results show that the TS-based methods have significantly lower regret than both UCB and BUCB. Similarly, the GP-based methods generally have lower regret than their BI-based counterparts. Thereby, GP-TS yields the best results in terms of minimizing cumulative regret. Finally, we observe that GP-BUCB has lower regret than GP-UCB. In the next experiment, we investigate how the parametrization of these two algoritms affects the results.

**BUCB parametrization**  As discussed in Section 2.2, GP-UCB and GP-BUCB differ mainly in their parametrization of the confidence parameter $\beta_t$. The confidence parameter determines the balance between exploration and exploitation. It is known that theoretical results tend to provide $\beta_t$ values that overexplore (Russo & Roy, 2014). Using the parameters of $\beta_t$ for GP-BUCB ($\omega$ and $\xi$), we can tune GP-BUCB towards more exploitation whilst retaining theoretical guarantees. We compare two theoretically valid choices of parametrizations for GP-BUCB ($\omega = 1, \xi = 1$ and $\omega = 1, \xi = 0.5$) against two parametrizations of GP-UCB where the first is theoretically valid and the

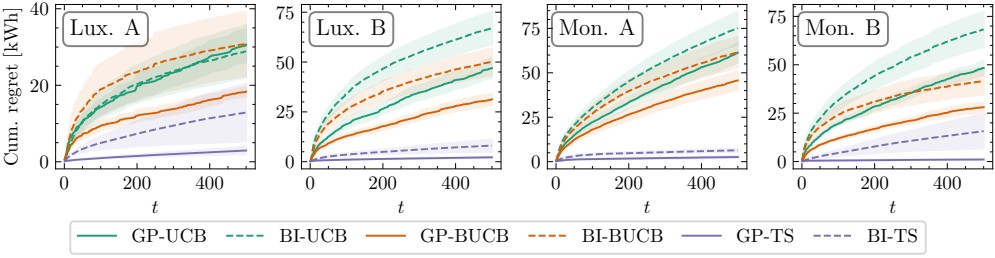

Figure 2: Cumulative regret for UCB, BUCB and TS using GP and Bayesian inference (BI) methods. The lines and regions correspond to the mean and $\pm 1$ standard error.

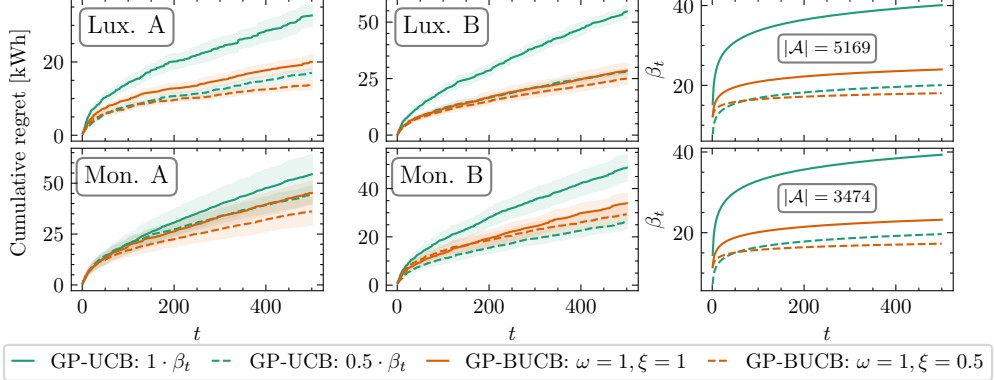

Figure 3: Cumulative regret of GP-UCB and GP-BUCB (left and middle column) for different parametrizations of $\beta_t$ (right column).

second has scaled $\beta_t$ by 0.5. The four parametrizations are evaluated 5 times each on the four routes in the Luxembourg and Monaco networks with a horizon of $T = 500$. The cumulative regret and the $\beta_t$ values are shown in Fig. 3. The theoretically valid $\beta_t$ values for GP-BUCB are smaller than for GP-UCB. By lowering $\xi$ from 1 to 0.5, the quantile parameter $\eta_t$ goes from $\mathcal{O}(t^{-1})$ to $\mathcal{O}(t^{-0.5})$ and using $\omega = 1$ the theoretial cumulative regret remains $\mathcal{O}(\sqrt{T})$.[2] The experimental results indicate that the parametrization with lower $\beta_t$ generally has lower cumulative regret. Using GP-BUCB, we gain more control of $\beta_t$ without sacrificing the theoretical guarantees.

**Impact of lengthscale** Finally, we investigate varying the kernel lengthscale to ensure our results are consistent and stable. A large lengthscale increases the correlation between edges, which should lower the regret of the GP-methods. Whilst a lower lengthscale decreases the correlation which should increase the regret of the GP-methods. We evaluate GP-BUCB and GP-TS against BI-BUCB and BI-TS with the kernel lengthscale varying between 0.1 and 2.0. Each combination of lengthscale and bandit-method is evaluated 5 times on all four routes with a horizon of $T = 500$. The final cumulative regret at $t = 500$ for the different lengthscales is shown in Figs. 4 and 5 in Appendix C.1. For GP-based methods, increasing the lengthscale increases the cumulative regret overall but for BI-based methods, there is no discernable pattern.

## 5 CONCLUSION

We presented novel Bayesian regret bounds for the combinatorial volatile Gaussian process semi-bandit for three GP-based bandit algorithms: GP-UCB, GP-BayesUCB and GP-TS. Additionally, we experimentally evaluated our contextual combinatorial GP method on the online energy-efficient navigation problem on real-world networks.

---

[2]Technically, one must use $\xi \leq 0.5 - \delta$ and $\omega \geq 1 + \delta$ for some $\delta > 0$. However, we could choose $\delta$ to be small enough such that GP-BUCB would select the exact same routes in all experiments.

ACKNOWLEDGMENTS

The work of Jack Sandberg and Morteza Haghir Chehreghani was partially supported by the Wallenberg AI, Autonomous Systems and Software Program (WASP) funded by the Knut and Alice Wallenberg Foundation. The work of Niklas Åkerblom was partially funded by the Strategic Vehicle Research and Innovation Programme (FFI) of Sweden, through the project EENE (reference number: 2018-01937). The computations were enabled by resources provided by the National Academic Infrastructure for Supercomputing in Sweden (NAISS), partially funded by the Swedish Research Council through grant agreement no. 2022-06725. Map data from Openstreetmap and available from `www.openstreetmap.org/copyright`.

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

# A    PROOFS

## A.1    FINITE CASE

In this section, we state and prove the regret bounds in the finite case for the three bandit algorithms GP-UCB, GP-BUCB and GP-TS. To begin, we establish lemmas that demonstrate the general procedure for the proofs and later we combine the lemmas to get the desired regret bounds.

In the following lemma, the Bayesian regret is separated into two terms.

**Lemma A.1.** *For GP-TS or any GP-UCB method the following upper bound on the Bayesian regret holds (with equality for GP-TS):*

$$BR(T) \leq \sum_{t \in [T]} \mathbb{E}[f(\mathbf{a}_t^*) - U_t(\mathbf{a}_t^*)] + \mathbb{E}[U_t(\mathbf{a}_t) - f(\mathbf{a}_t)]. \tag{9}$$

*Proof.* The proof follows the procedure of Prop. 1 by Russo & Roy (2014) for GP-TS and Thm. B.1. by Takeno et al. (2023) for GP-UCB. For GP-TS,

$$\mathrm{BR}(T) = \sum_{t \in [T]} \mathbb{E}\left[f(\mathbf{a}_t^*) - f(\mathbf{a}_t)\right] \tag{10}$$

$$= \sum_{t \in [T]} \mathbb{E}_{H_t}\left[\mathbb{E}_t\left[f(\mathbf{a}_t^*) - U_t(\mathbf{a}_t^*) + U_t(\mathbf{a}_t) - f(\mathbf{a}_t)|H_t\right]\right] \qquad \left(\mathbf{a}_t^*|H_t \overset{d}{=} \mathbf{a}_t|H_t\right) \tag{11}$$

$$= \sum_{t \in [T]} \mathbb{E}[f(\mathbf{a}_t^*) - U_t(\mathbf{a}_t^*)] + \sum_{t \in [T]} \mathbb{E}[U_t(\mathbf{a}_t) - f(\mathbf{a}_t)]. \tag{12}$$

Similarly, for any GP-UCB method,

$$\mathrm{BR}(T) = \sum_{t \in [T]} \mathbb{E}\left[f(\mathbf{a}_t^*) - f(\mathbf{a}_t)\right] \tag{13}$$

$$= \sum_{t \in [T]} \mathbb{E}\left[f(\mathbf{a}_t^*) - U_t(\mathbf{a}_t^*) + U_t(\mathbf{a}_t^*) - U_t(\mathbf{a}_t) + U_t(\mathbf{a}_t) - f(\mathbf{a}_t)\right] \tag{14}$$

$$\leq \sum_{t \in [T]} \mathbb{E}\left[f(\mathbf{a}_t^*) - U_t(\mathbf{a}_t^*) + U_t(\mathbf{a}_t) - f(\mathbf{a}_t)\right] \tag{15}$$

where the final step uses that $U_t(\mathbf{a}_t^*) - U_t(\mathbf{a}_t) \leq 0$ since $\mathbf{a}_t = \arg\max_{\mathbf{a} \in \mathcal{S}_t} U_t(\mathbf{a})$. $\qquad\square$

Whilst Lemma A.1 applies to all the considered bandit algorithms, the two terms in the decomposition requires knowing the specific bandit algorithm. Bounding the left term requires knowledge of the confidence parameter $\beta_t$. Therefore we present a lemma that applies to GP-UCB and GP-TS, and another lemma that applies to GP-BUCB.

**Lemma A.2.** *If $|\mathcal{A}| < \infty$, then*

$$\sum_{t \in [T]} \mathbb{E}[f(\mathbf{a}_t^*) - U_t(\mathbf{a}_t^*)] \leq \frac{\pi^2}{6} \tag{16}$$

*holds for GP-UCB and GP-TS with $\beta_t = 2\log\left(|\mathcal{A}|t^2/\sqrt{2\pi}\right)$.*

*Proof.* The proof closely follows the proof of Thm. B.1 by Takeno et al. (2023). Let $R_1 = \sum_{t \in [T]} \mathbb{E}[f(\mathbf{a}_t^*) - U_t(\mathbf{a}_t^*)]$, then

$$R_1 = \sum_{t \in [T]} \mathbb{E}_{H_t} \left[ \mathbb{E}_t \left[ f(\mathbf{a}_t^*) - U_t(\mathbf{a}_t^*) | H_t \right] \right] \tag{17}$$

$$= \sum_{t \in [T]} \mathbb{E}_{H_t} \left[ \mathbb{E}_t \left[ \sum_{a \in \mathbf{a}_t^*} f(a) - U_t(a) \bigg| H_t \right] \right] \tag{18}$$

$$\leq \sum_{t \in [T]} \mathbb{E}_{H_t} \left[ \mathbb{E}_t \left[ \sum_{a \in \mathbf{a}_t^*} (f(a) - U_t(a))_+ \bigg| H_t \right] \right] \qquad ((x)_+ := \max(0, x) \geq x) \tag{19}$$

$$\leq \sum_{t \in [T]} \mathbb{E}_{H_t} \left[ \sum_{a \in \mathcal{A}} \mathbb{E}_t \left[ (f(a) - U_t(a))_+ \bigg| H_t \right] \right]. \qquad (\mathbf{a}_t^* \subseteq \mathcal{A}) \tag{20}$$

Note that $f(a) - U_t(a) | H_t \sim \mathcal{N}(-\sqrt{\beta_t} \sigma_{t-1}(a), \sigma_{t-1}^2(a))$. As Russo & Roy (2014), by using that if $X \sim \mathcal{N}(\mu, \sigma^2)$ for $\mu \leq 0$, then $\mathbb{E}[(X)_+] \leq \frac{\sigma}{\sqrt{2\pi}} \exp\left(\frac{-\mu^2}{2\sigma^2}\right)$, we get the following for $R_1$:

$$R_1 \leq \sum_{t \in [T]} \mathbb{E}_{H_t} \left[ \sum_{a \in \mathcal{A}} \mathbb{E}_t \left[ \frac{\sigma_{t-1}(a)}{\sqrt{2\pi}} \exp\left(\frac{-\beta_t}{2}\right) \bigg| H_t \right] \right] \tag{21}$$

$$\leq \sum_{t \in [T]} \frac{|\mathcal{A}|}{\sqrt{2\pi}} \exp\left(\frac{-\beta_t}{2}\right) \qquad \left(\sigma_{t-1}^2(a) \leq k(a, a) \leq 1\right) \tag{22}$$

$$\leq \sum_{t \in [T]} \frac{1}{t^2} \qquad \left(\beta_t = 2 \log(|\mathcal{A}| t^2 / \sqrt{2\pi}\right) \tag{23}$$

$$\leq \frac{\pi^2}{6}. \qquad \left(\sum_{t=1}^{\infty} \frac{1}{t^2} = \frac{\pi^2}{6}\right) \tag{24}$$

$\square$

**Lemma 3.1.** *Let* $C_\omega = \left(\sqrt{\pi}\omega / \sqrt{2e(\omega - 1)}\right)^{1/\omega}$, *then for GP-BUCB with confidence parameter* $\beta_t = 2\left(\mathrm{erf}^{-1}(1 - 2\eta_t)\right)^2$ *and* $\eta_t = \frac{\sqrt{2\pi}\omega}{2|\mathcal{A}|^\omega t^\xi}$, $\xi > 0$, $\omega > 1$,

$$\sum_{t \in [T]} \mathbb{E}[f(\mathbf{a}_t^*) - U_t(\mathbf{a}_t^*)] \leq C_\omega \cdot \begin{cases} \frac{\omega}{\omega - \xi} T^{1 - \frac{\xi}{\omega}} & \text{if } \xi/\omega < 1, \\ 1 + \log T & \text{if } \xi/\omega = 1, \\ \frac{\xi}{\xi - \omega} & \text{if } \xi/\omega > 1. \end{cases}$$

*Proof.* Following the proof of Lemma A.2, we get that

$$\sum_{t \in [T]} \mathbb{E}\left[f(\mathbf{a}_t^*) - U_t(\mathbf{a}_t^*)\right] \leq \sum_{t \in [T]} \frac{|\mathcal{A}|}{\sqrt{2\pi}} \exp\left(-\frac{\beta_t}{2}\right). \tag{25}$$

Note that, according to Lemma A.13, $\mathrm{erf}^{-1}(u) \geq \sqrt{-\omega^{-1} \log((1 - u)/\vartheta)}$ for $\omega > 1$ and $\vartheta = \sqrt{2e/\pi}\sqrt{\omega - 1}/\omega$. We use the largest value of $\vartheta$ permitted by Lemma A.13 since it yields the

tightest bound. Then,

$$\sum_{t \in [T]} \frac{|\mathcal{A}|}{\sqrt{2\pi}} \exp\left(-\frac{\beta_t}{2}\right) = \sum_{t \in [T]} \frac{|\mathcal{A}|}{\sqrt{2\pi}} \exp\left(-\left(\text{erf}^{-1}(1 - 2\eta_t)\right)^2\right) \tag{26}$$

$$\leq \sum_{t \in [T]} \frac{|\mathcal{A}|}{\sqrt{2\pi}} \exp\left(\omega^{-1} \log\left(\frac{1 - (1 - 2\eta_t)}{\vartheta}\right)\right) \quad \text{(Lemma A.13)} \tag{27}$$

$$= \sum_{t \in [T]} \frac{|\mathcal{A}|}{\sqrt{2\pi}} \left(\frac{2\eta_t}{\vartheta}\right)^{\frac{1}{\omega}} \tag{28}$$

$$= \sum_{t \in [T]} \vartheta^{-\frac{1}{\omega}} t^{-\frac{\xi}{\omega}} \quad (\text{Def. of } \eta_t) \tag{29}$$

$$= \left(\frac{\sqrt{\pi}\omega}{\sqrt{2e(\omega - 1)}}\right)^{\frac{1}{\omega}} \sum_{t \in [T]} t^{-\frac{\xi}{\omega}}. \quad (\text{Def. of } \vartheta) \tag{30}$$

The behaviour of $\sum_{t \in [T]} t^{-\frac{\xi}{\omega}}$ critically depends on the ratio $\xi/\omega$. First, if $\xi/\omega < 1$, then

$$\sum_{t \in [T]} t^{-\frac{\xi}{\omega}} \leq \int_0^T t^{-\frac{\xi}{\omega}} \mathrm{d}t = T^{1-\frac{\xi}{\omega}} \frac{1}{1 - \frac{\xi}{\omega}}. \tag{31}$$

Second, if $\xi/\omega = 1$, then

$$\sum_{t \in [T]} t^{-1} \leq 1 + \int_1^T t^{-1} dt = 1 + \log T. \tag{32}$$

Finally, if $\xi/\omega > 1$, then

$$\sum_{t \in [T]} t^{-\frac{\xi}{\omega}} \leq 1 + \int_1^\infty t^{-\frac{\xi}{\omega}} \mathrm{d}t = 1 + \left[\frac{1}{1 - \frac{\xi}{\omega}} t^{1 - \frac{\xi}{\omega}}\right]_1^\infty = 1 - \frac{1}{1 - \frac{\xi}{\omega}} = \frac{\xi}{\xi - \omega}. \tag{33}$$

$\square$

Before we bound the right term in Lemma A.1, we introduce a lemma for the confidence radius that applies to all the bandit algorithms considered.

**Lemma A.3.**

$$\sum_{t \in [T]} \mathbb{E}\left[\sum_{a \in \mathbf{a}_t} \sqrt{\beta_t} \sigma_{t-1}(a)\right] \leq \sqrt{2(\lambda_K^* + \varsigma^2) T K \beta_T \gamma_{TK}} \tag{34}$$

*for GP-TS or any GP-UCB method with increasing confidence parameter $\beta_t$.*

*Proof.*

$$\sum_{t \in [T]} \mathbb{E} \left[ \sum_{a \in \mathbf{a}_t} \sqrt{\beta_t} \sigma_{t-1}(a) \right] \tag{35}$$

$$= \mathbb{E} \left[ \sum_{t \in [T]} \sum_{a \in \mathbf{a}_t} \sqrt{\beta_t} \sigma_{t-1}(a) \right] \tag{36}$$

$$\leq \mathbb{E} \left[ \sqrt{\sum_{t \in [T]} \sum_{a \in \mathbf{a}_t} \beta_t} \sqrt{\sum_{t \in [T]} \sum_{a \in \mathbf{a}_t} \sigma_{t-1}^2(a)} \right] \qquad \text{(Cauchy-Schwarz inequality)} \tag{37}$$

$$\leq \mathbb{E} \left[ \sqrt{TK\beta_T} \sqrt{\sum_{t \in [T]} \sum_{a \in \mathbf{a}_t} \sigma_{t-1}^2(a)} \right] \qquad \left( |\mathbf{a}_t| \leq K, \max_{t \in [T]} \beta_t = \beta_T \right) \tag{38}$$

$$= \sqrt{TK\beta_T} \mathbb{E} \left[ \sqrt{\sum_{t \in [T]} \sum_{a \in \mathbf{a}_t} \sigma_{t-1}^2(a)} \right] \tag{39}$$

$$\leq \sqrt{TK\beta_T} \mathbb{E} \left[ \sqrt{2(\lambda_K^* + \varsigma^2)\gamma_{TK}} \right] \qquad \text{(Lemma A.12)} \tag{40}$$

$$\leq \sqrt{2(\lambda_K^* + \varsigma^2)TK\beta_T\gamma_{TK}}. \tag{41}$$

$\square$

Next, we show how the right term in Lemma A.1 can be rewritten in terms of the confidence radius for any GP-UCB method.

**Lemma A.4.**

$$\sum_{t \in [T]} \mathbb{E} \left[ U_t(\mathbf{a}_t) - f(\mathbf{a}_t) \right] = \sum_{t \in [T]} \mathbb{E} \left[ \sqrt{\beta_t} \sigma_{t-1}(\mathbf{a}_t) \right] \tag{42}$$

*for any GP-UCB method with confidence parameter $\beta_t$.*

*Proof.* Note that given the history $H_t$, $\mathbf{a}_t := \arg\max_{\mathbf{a} \in \mathcal{S}_t} U_t(\mathbf{a})$ is deterministic. Thus,

$$\sum_{t \in [T]} \mathbb{E} \left[ U_t(\mathbf{a}_t) - f(\mathbf{a}_t) \right] = \sum_{t \in [T]} \mathbb{E}_{H_t} \left[ \mathbb{E}_t \left[ U_t(\mathbf{a}_t) - f(\mathbf{a}_t) | H_t \right] \right] \tag{43}$$

$$= \sum_{t \in [T]} \mathbb{E}_{H_t} \left[ \mathbb{E}_t \left[ \mu_{t-1}(\mathbf{a}_t) + \sqrt{\beta_t} \sigma_{t-1}(\mathbf{a}_t) - f(\mathbf{a}_t) \Big| H_t \right] \right] \tag{44}$$

$$= \sum_{t \in [T]} \mathbb{E}_{H_t} \left[ \mathbb{E}_t \left[ \mu_{t-1}(\mathbf{a}_t) + \sqrt{\beta_t} \sigma_{t-1}(\mathbf{a}_t) - \mu_{t-1}(\mathbf{a}_t) \Big| H_t \right] \right] \tag{45}$$

$$= \sum_{t \in [T]} \mathbb{E} \left[ \sqrt{\beta_t} \sigma_{t-1}(\mathbf{a}_t) \right]. \tag{46}$$

$\square$

For the final lemma in the finite case, we bound the right term in Lemma A.1 for Thompson sampling using the previous results.

**Lemma A.5.**

$$\sum_{t \in [T]} \mathbb{E} \left[ U_t(\mathbf{a}_t) - f(\mathbf{a}_t) \right] \leq 2\sqrt{C_K TK \beta_T \gamma_{TK}} + \frac{\pi^2}{6} \tag{47}$$

*holds for GP-TS with $\beta_t = 2\log\left(|\mathcal{A}|t^2/\sqrt{2\pi}\right)$.*

*Proof.* By adding and subtracting the lower bound $L(\mathbf{a}_t)$, we obtain

$$\sum_{t \in [T]} \mathbb{E}\left[U_t(\mathbf{a}_t) - f(\mathbf{a}_t)\right] = \sum_{t \in [T]} \mathbb{E}\left[U_t(\mathbf{a}_t) - f(\mathbf{a}_t) + L(\mathbf{a}_t) - L(\mathbf{a}_t)\right] \tag{48}$$

$$= \sum_{t \in [T]} \mathbb{E}\left[U_t(\mathbf{a}_t) - L(\mathbf{a}_t)\right] + \sum_{t \in [T]} \mathbb{E}\left[L(\mathbf{a}_t) - f(\mathbf{a}_t)\right] \tag{49}$$

$$= 2\underbrace{\sum_{t \in [T]} \mathbb{E}\left[\sqrt{\beta_t}\sigma_{t-1}(\mathbf{a}_t)\right]}_{(1)} + \underbrace{\sum_{t \in [T]} \mathbb{E}\left[L(\mathbf{a}_t) - f(\mathbf{a}_t)\right]}_{(2)}. \tag{50}$$

By Lemma A.3, $(1) \le \sqrt{2(\lambda_K^* + \varsigma^2)TK\beta_T\gamma_{TK}}$. The bound $(2) \le \frac{\pi^2}{6}$ is obtained using the same steps as in Lemma A.2 due to the symmetry of $L(\mathbf{a}_t) - f(\mathbf{a}_t)$ and $f(\mathbf{a}_t) - U(\mathbf{a}_t)$. $\qquad\square$

Finally, we present and prove the regret bounds for GP-UCB, GP-BUCB and GP-TS using the established lemmas.

**Theorem 3.2** (Finite regret bounds). *Let $C_K := 2(\lambda_K^* + \varsigma^2)$. When $\mathcal{A}$ is finite, the Bayesian regret of*

   *(i) GP-UCB with $\beta_t = 2\log(|\mathcal{A}|t^2/\sqrt{2\pi})$ is bounded as $BR(T) \le \frac{\pi^2}{6} + \sqrt{C_K TK\beta_T\gamma_{TK}}$.*

   *(ii) GP-BUCB with $\beta_t = 2\left(\mathrm{erf}^{-1}(1 - 2\eta_t)\right)^2$ for $\eta_t = \frac{\sqrt{2\pi}^\omega}{2|\mathcal{A}|^\omega t^\xi}$, $\xi > \omega > 1$ is bounded as $BR(T) \le \sqrt{C_K TK\beta_T\gamma_{TK}} + C_\omega \cdot \frac{\xi}{\xi - \omega}$ where $C_\omega = \left(\sqrt{\pi}\omega/\sqrt{2e(\omega - 1)}\right)^{1/\omega}$.*

   *(iii) GP-TS is bounded as $BR(T) \le \frac{\pi^2}{3} + 2\sqrt{C_K TK\beta_T\gamma_{TK}}$ where $\beta_t = 2\log\left(|\mathcal{A}|t^2/\sqrt{2\pi}\right)$.*

*Proof.* (i) The regret bound for GP-UCB is obtained as follows:

$$BR(T) \le \sum_{t \in [T]} \mathbb{E}\left[f(\mathbf{a}_t^*) - U_t(\mathbf{a}_t^*)\right] + \sum_{t \in [T]} \mathbb{E}\left[U_t(\mathbf{a}_t) - f(\mathbf{a}_t)\right] \qquad \text{(Lemma A.1)} \tag{51}$$

$$\le \frac{\pi^2}{6} + \sum_{t \in [T]} \mathbb{E}\left[\sqrt{\beta_t}\sigma_{t-1}(\mathbf{a}_t)\right] \qquad \text{(Lemmas A.2 and A.4)} \tag{52}$$

$$\le \frac{\pi^2}{6} + \sqrt{2(\lambda_K^* + \varsigma^2)TK\beta_T\gamma_{TK}}. \qquad \text{(Lemma A.3)} \tag{53}$$

(ii) The regret of GP-BUCB can be decomposed as follows:

$$BR(T) \le \sum_{t \in [T]} \mathbb{E}\left[U_t(\mathbf{a}_t) - f(\mathbf{a}_t)\right] + \sum_{t \in [T]} \mathbb{E}\left[f(\mathbf{a}_t^*) - U_t(\mathbf{a}_t^*)\right] \qquad \text{(Lemma A.1)} \tag{54}$$

$$\le \sum_{t \in [T]} \mathbb{E}\left[\sqrt{\beta_t}\sigma_{t-1}(\mathbf{a}_t)\right] \qquad \text{(Lemma A.4)} \tag{55}$$

$$+ \left(\frac{\sqrt{\pi}\omega}{\sqrt{2e(\omega - 1)}}\right)^{1/\omega} \cdot \begin{cases} \frac{\omega}{\omega - \xi}T^{1 - \frac{\xi}{\omega}} & \text{if } \xi/\omega < 1, \\ \frac{\xi}{\xi - \omega} & \text{if } \xi/\omega > 1. \end{cases} \qquad \text{(Lemma 3.1)} \tag{56}$$

From Lemma A.3, $\sum_{t \in [T]} \mathbb{E}\left[\sqrt{\beta_t}\sigma_{t-1}(\mathbf{a}_t)\right] \le \sqrt{2(\lambda_K^* + \varsigma^2)TK\beta_T\gamma_{TK}}$ and we obtain the desired result.

(iii) The regret of GP-TS is obtained as follows:

$$BR(T) = \sum_{t \in [T]} \mathbb{E}\left[f(\mathbf{a}_t^*) - U_t(\mathbf{a}_t^*)\right] + \sum_{t \in [T]} \mathbb{E}\left[U_t(\mathbf{a}_t) - f(\mathbf{a}_t)\right] \qquad \text{(Lemma A.1)} \tag{57}$$

$$\le \frac{\pi^2}{6} + \frac{\pi^2}{6} + 2\sqrt{C_K TK\beta_T\gamma_{TK}}. \qquad \text{(Lemmas A.2 and A.5)} \tag{58}$$

$\qquad\square$

## A.2 INFINITE CASE

Similar to the finite case, we establish lemmas that hold for all bandit algorithms and finally state and prove the regret bounds.

Before stating the first lemma, we restate the assumptions for convenience:

**Assumption 3.3** (Regularity assumptions). *Assume $\mathcal{A} \subset [0, C_1]^d$ is a compact and convex set for some $C_1 > 0$. Furthermore, assume that $\mu$ and $k$ are both L-Lipschitz on $\mathcal{A}$ and $\mathcal{A} \times \mathcal{A}$, respectively, for some $L > 0$. In addition, for $f \sim \mathcal{GP}(\mu, k)$ assume that there exists constants $C_2, C_3 > 0$ such that:*

$$\mathbb{P}\left(\sup_{a \in \mathcal{A}} \left| \frac{\partial f}{\partial a^{(j)}} \right| > l \right) \leq C_2 \exp\left(-\frac{l^2}{C_3^2}\right), \tag{5}$$

*for $j \in \{1, \ldots, d\}$ and $l > 0$ where $a^{(j)}$ denotes the $j$-th element of $a$.*

**Remark A.6.** *By Thm. 5 of Ghosal & Roy (2006), the high probability bound holds if $\mu$ is continuously differentiable and $k$ is 4 times differentiable, which would also imply the Lipschitzness of $\mu$ and $k$. As discussed by Srinivas et al. (2012), this holds for the Matérn kernel if $\nu \geq 2$ by a result of Stein (1999) and holds trivially for the squared exponential kernel. Thus, the Lipschitz assumption of $\mu$ and $k$ is not particularly restrictive.*

**Assumption 3.4** (Discretization size). *Let $\tau_t$ denote the number of discretization points per dimension and assume that*

$$\begin{cases} \tau_t \geq 2t^2 K L d C_1 (1 + tK\varsigma^{-1}), & \text{(6a)} \\ \tau_t/\beta_t \geq 8t^4 K^2 L d C_1, & \text{(6b)} \\ \tau_t^2/\beta_t \geq 8t^5 K^3 L^2 d^2 C_1^2 \varsigma^{-2}, & \text{(6c)} \\ \tau_t \geq t^2 K d C_1 C_3 (\sqrt{\log(C_2 d)} + \sqrt{\pi}/2) & \text{(6d)} \end{cases}$$

*where the constants $C_1, C_2, C_3$ and $L$ are given by Assumption 3.3 whilst the constants $d, K$ and $\varsigma$ are defined by the bandit problem (Section 2.1).*

**Remark A.7.** *For the theorems to be relevant, the assumptions imposed on $\tau_t$ must be satisfiable for some $\tau_t$. If $\beta_t = 2\log\left(\frac{\tau_t^d t^2}{\sqrt{2\pi}}\right)$, then Assumption 3.4 is satisfied by*

$$\tau_t = \max \begin{cases} 2K L d C_1 (1 + tK\varsigma^{-1})t^2, \\ \left(\left(16t^4 K^2 L d C_1\right)\left(d + \log\left(\frac{t^2}{\sqrt{2\pi}}\right)\right)\right)^{\frac{1}{1-1/e}}, \\ \left(\left(16t^5 K^3 L^2 d^2 C_1^2 \varsigma^{-2}\right)\left(d + \log\left(\frac{t^2}{\sqrt{2\pi}}\right)\right)\right)^{\frac{1}{2-1/e}}, \\ t^2 K d C_1 C_3 \left(\sqrt{\log(C_2 d)} + \frac{\sqrt{\pi}}{2}\right). \end{cases} \tag{59}$$

*This can be shown by noting that $\log \tau_t \leq \sqrt[e]{\tau_t}$ and $1 \leq \sqrt[e]{\tau_t}$ and then deriving that $\frac{1}{\beta_t} \geq \frac{1}{\tau_t^{1/e}(d+\log(t^2/\sqrt{2\pi}))}$.*

*Similarly, if $\beta_t = 2\left(\text{erf}^{-1}(1 - 2\eta_t)\right)^2$ and $\eta_t = \frac{\sqrt{2\pi}^\omega}{2\tau_t^{d\omega} t^\xi}$, $\omega > 1$, then Assumption 3.4 is satisfied by*

$$\tau_t = \max \begin{cases} 2t^2 K L d C_1 (1 + tK\varsigma^{-1}), \\ \left(\left(16t^4 K^2 L d C_1\right)\left(d\omega + \log\left(\frac{t^\xi}{2\sqrt{2\pi}^\omega}\right)\right)\right)^{\frac{1}{1-1/e}}, \\ \left(\left(16t^5 K^3 L^2 d^2 C_1^2 \varsigma^{-2}\right)\left(d\omega + \log\left(\frac{t^\xi}{2\sqrt{2\pi}^\omega}\right)\right)\right)^{\frac{1}{2-1/e}}, \\ t^2 K d C_1 C_3 \left(\sqrt{\log(C_2 d)} + \frac{\sqrt{\pi}}{2}\right). \end{cases} \tag{60}$$

*This is shown similarly as before but using Lemma A.14 to upper bound $\text{erf}^{-1}(1 - 2\eta_t)$ in $\beta_t$.*

Next, we present a lemma that bounds the discretization error of the expected reward of optimal super arm.

**Lemma A.8.** *Let $\mathcal{D}_t \subset \mathcal{A}$ be a finite discretization with each dimension equally divided into $\tau_t = t^2 K d C_1 C_3 \left( \sqrt{\log(C_2 d)} + \sqrt{\pi}/2 \right)$ such that $|\mathcal{D}_t| = \tau_t^d$. Then,*

$$\sum_{t \in [T]} \mathbb{E}\left[ f(\mathbf{a}_t^*) - f([\mathbf{a}_t^*]_{\mathcal{D}_t}) \right] \leq \frac{\pi^2}{6}. \tag{61}$$

*Proof.*

$$\sum_{t \in [T]} \mathbb{E}[f(\mathbf{a}_t^*) - f([\mathbf{a}_t^*]_{\mathcal{D}_t})] = \sum_{t \in [T]} \mathbb{E}\left[ \sum_{a \in \mathbf{a}_t^*} f(a) - f([a]_{\mathcal{D}_t}) \right] \tag{62}$$

$$\leq K \sum_{t \in [T]} \mathbb{E}\left[ \sup_{a \in \mathcal{A}} f(a) - f([a]_{\mathcal{D}_t}) \right] \quad (|\mathbf{a}_t^*| \leq K) \tag{63}$$

$$\leq K \sum_{t \in [T]} \frac{1}{Kt^2} \quad \begin{pmatrix} \text{Lemma} & \text{H.2} & \text{of} \\ \text{Takeno et al. (2023)} \\ \text{with } u_t = Kt^2 \end{pmatrix} \tag{64}$$

$$\leq \frac{\pi^2}{6} \quad \left( \sum_{t=1}^{\infty} \frac{1}{t^2} = \frac{\pi^2}{6} \right) \tag{65}$$

$\square$

In the following lemma, we bound the discretization error of the posterior mean and standard deviation in terms of the regularity parameters, the discretization size and number of arms selected.

**Lemma A.9.** *Let $\mu_{t-1}$ and $\sigma_{t-1}$ denote the posterior mean and standard deviation of $\mathcal{GP}(\mu, k)$ after sampling $N_{t-1}$ base arms. If $a \in \mathcal{A}$, then*

$$\mu_{t-1}([a]_{\mathcal{D}_t}) - \mu_{t-1}(a) \leq L\frac{dC_1}{\tau_t} + L\frac{dC_1}{\tau_t} \sqrt{N_{t-1}} \varsigma^{-1} \sqrt{\|\mathbf{L}^{-1}(\mathbf{y} - \boldsymbol{\mu})\|_2^2} \tag{66}$$

*and*

$$\sigma_{t-1}([a]_{\mathcal{D}_t}) - \sigma_{t-1}(a) \leq \sqrt{L\frac{dC_1}{\tau_t} + N_{t-1}L^2 \left(\frac{dC_1}{\tau_t}\right)^2 \varsigma^{-2}} \tag{67}$$

*for $L-$Lipschitz $\mu$ and $k$ where $\mathbf{L}$ is the Cholesky decomposition of $\mathbf{K} + \varsigma^2 I$ and $\|\mathbf{L}^{-1}(\mathbf{y} - \boldsymbol{\mu})\|_2^2 \sim \chi^2$ with $N_{t-1}$ degrees of freedom.*

*Proof.* Consider first the difference in posterior mean:

$$\mu_{t-1}([a]_{\mathcal{D}_t}) - \mu_{t-1}(a) \tag{68}$$

$$= \mu([a]_{\mathcal{D}_t}) - \mu(a) + (\mathbf{k}([a]_{\mathcal{D}_t}) - \mathbf{k}(a))^\top \left(\mathbf{K} + \varsigma^2 I\right)^{-1} (\mathbf{y} - \boldsymbol{\mu}) \tag{69}$$

$$\leq L \sup_{a \in \mathcal{A}} \|a - [a]_{\mathcal{D}_t}\|_1 + \left\| (\mathbf{k}([a]_{\mathcal{D}_t}) - \mathbf{k}(a))^\top (\mathbf{K} + \varsigma^2 I)^{-1}(\mathbf{y} - \boldsymbol{\mu}) \right\|_2 \quad (\mu \ L\text{-Lipschitz}) \tag{70}$$

$$\leq L\frac{dC_1}{\tau_t} + \left\| (\mathbf{k}([a]_{\mathcal{D}_t}) - \mathbf{k}(a))^\top (\mathbf{K} + \varsigma^2 I)^{-1}(\mathbf{y} - \boldsymbol{\mu}) \right\|_2 \tag{71}$$

where the last step uses that $\sup_{a \in \mathcal{A}} \|a - [a]_{\mathcal{D}_t}\|_1 \leq \frac{dC_1}{\tau_t}$.

Next, we will appropriately split the norm into a product of norms and bound the individual factors. Let $\mathbf{K} + \varsigma^2 I = \mathbf{L}\mathbf{L}^\top$ denote the Cholesky decomposition. Note that $\mathbf{y} - \boldsymbol{\mu} \sim \mathcal{N}(0, \mathbf{K} + \varsigma^2 I)$. Then, $\mathbf{L}^{-1}(\mathbf{y} - \boldsymbol{\mu}) \sim \mathcal{N}(0, \mathbf{L}^{-1}\mathbf{L}\mathbf{L}^\top(\mathbf{L}^{-1})^\top) = \mathcal{N}(0, I)$ and thus $\|\mathbf{L}^{-1}(\mathbf{y} - \boldsymbol{\mu})\|_2$ has a chi distribution with $N_{t-1}$ degrees of freedom.

Let $\mathrm{eig}(A)$ denote the set of eigenvalues of the square matrix $A$. The matrix norm of the inverted Cholesky decomposition $\mathbf{L}^{-1}$ can be bounded as:

$$\|\mathbf{L}^{-1}\|_2 = \sqrt{\max \mathrm{eig}\left((\mathbf{L}^{-1})^\top \mathbf{L}^{-1}\right)} \qquad \begin{pmatrix} \text{Eq. (538) of Petersen \&} \\ \text{Pedersen (2012)} \end{pmatrix} \tag{72}$$

$$= \sqrt{\max \mathrm{eig}\left((\mathbf{K} + \varsigma^2 I)^{-1}\right)} \tag{73}$$

$$= \sqrt{\max \frac{1}{\mathrm{eig}\left(\mathbf{K} + \varsigma^2 I\right)}} \tag{74}$$

$$= \sqrt{\max \frac{1}{\mathrm{eig}\left(\mathbf{K}\right) + \varsigma^2}} \leq \sqrt{\frac{1}{\varsigma^2}} \leq \frac{1}{\varsigma}. \qquad (\mathbf{K}\ \text{p.s.d.}, \varsigma > 0) \tag{75}$$

Similarly, we also get that

$$\|(\mathbf{K} + \varsigma^2 I)^{-1}\|_2 \leq \varsigma^{-2}. \tag{76}$$

The kernel difference can be bounded as follows:

$$\|\mathbf{k}([a]_{\mathcal{D}_t}) - \mathbf{k}(a)\|_2 = \sqrt{\sum_{i=1}^{N_{t-1}} \left(k([a]_{\mathcal{D}_t}, x_i) - k(a, x_i)\right)^2} \tag{77}$$

$$\leq \sqrt{\sum_{i=1}^{N_{t-1}} L^2 \left(\frac{dC_1}{\tau_t}\right)^2} \leq L \frac{dC_1}{\tau_t} \sqrt{N_{t-1}} \tag{78}$$

where we use the fact that $k$ is $L$-Lipschitz. Applying Cauchy-Schwarz and the obtained bounds, we find that

$$\mu_{t-1}([a]_{\mathcal{D}_t}) - \mu_{t-1}(a) \leq L \frac{dC_1}{\tau_t} + L \frac{dC_1}{\tau_t} \sqrt{N_{t-1}} \varsigma^{-1} \|\mathbf{L}^{-1}(\mathbf{y} - \boldsymbol{\mu})\|_2. \tag{79}$$

The posterior standard deviation is bounded similarly:

$$\sigma_{t-1}([a]_{\mathcal{D}_t}) - \sigma_{t-1}(a) \leq \sqrt{\left|\sigma_{t-1}^2([a]_{\mathcal{D}_t}) - \sigma_{t-1}^2(a)\right|}. \tag{80}$$

Continuing,

$$\left|\sigma_{t-1}^2([a]_{\mathcal{D}_t}) - \sigma_{t-1}^2(a)\right| \tag{81}$$

$$= \left|k([a]_{\mathcal{D}_t}, [a]_{\mathcal{D}_t}) - k(a, a) + (\mathbf{k}([a]_{\mathcal{D}_t}) - \mathbf{k}(a))^\top \left(\mathbf{K} + \varsigma^2 I\right)^{-1} (\mathbf{k}([a]_{\mathcal{D}_t}) - \mathbf{k}(a))\right| \tag{82}$$

$$\leq |k([a]_{\mathcal{D}_t}, [a]_{\mathcal{D}_t}) - k(a, a)| + \left|(\mathbf{k}([a]_{\mathcal{D}_t}) - \mathbf{k}(a))^\top \left(\mathbf{K} + \varsigma^2 I\right)^{-1} (\mathbf{k}([a]_{\mathcal{D}_t}) - \mathbf{k}(a))\right| \tag{83}$$

$$\leq L \frac{dC_1}{\tau_t} + \|\mathbf{k}([a]_{\mathcal{D}_t}) - \mathbf{k}(a)\|_2^2 \left\|(\mathbf{K} + \varsigma^2 I)^{-1}\right\|_2 \tag{84}$$

$$\leq L \frac{dC_1}{\tau_t} + \left(L \frac{dC_1}{\tau_t} \sqrt{N_{t-1}}\right)^2 \varsigma^{-2}. \qquad (\text{Eqs. (76) and (78)}) \tag{85}$$

Combining the above, the final bound is:

$$\sigma_{t-1}([a]_{\mathcal{D}_t}) - \sigma_{t-1}(a) \leq \sqrt{L \frac{dC_1}{\tau_t} + N_{t-1} \left(L \frac{dC_2}{\tau_t}\right)^2 \varsigma^{-2}}. \tag{86}$$

$$\square$$

Using Lemma A.9, we are ready to construct a constant bound for the expected discretization error of the posterior mean:

**Lemma A.10.** *If Assumption 3.3 holds and $\tau_t$ satisfies Eq. (6a) in Assumption 3.4, then for any sequence of super arms $\mathbf{a}_t \in \mathcal{S}_t$ $t \geq 1$, the posterior mean $\mu_{t-1}(\mathbf{a})$ satisfies*

$$\sum_{t \in [T]} \mathbb{E}\left[\mu_{t-1}([\mathbf{a}_t]_{\mathcal{D}_t}) - \mu_{t-1}(\mathbf{a}_t)\right] \leq \frac{\pi^2}{12}. \tag{87}$$

*Proof.* Note that the assumption on $\tau_t$ is equivalent to $KL\frac{dC_1}{\tau_t}(1 + tK\varsigma^{-1}) \leq \frac{1}{2t^2}$. Then, we can bound the discretization error of the posterior mean as follows:

$$\sum_{t \in [T]} \mathbb{E}\left[\sum_{a \in \mathbf{a}_t} \mu_{t-1}([a]_{\mathcal{D}_t}) - \mu_{t-1}(a)\right] \tag{88}$$

$$\leq \sum_{t \in [T]} \mathbb{E}\left[K \sup_{a \in \mathcal{A}} [\mu_{t-1}([a]_{\mathcal{D}_t}) - \mu_{t-1}(a)]\right] \qquad (|\mathbf{a}_t| \leq K) \tag{89}$$

$$\leq \sum_{t \in [T]} \mathbb{E}\left[K \sup_{a \in \mathcal{A}} L\frac{dC_1}{\tau_t}\left(1 + \sqrt{tK}\varsigma^{-1}\sqrt{\|\mathbf{L}^{-1}(\mathbf{y} - \boldsymbol{\mu})\|_2^2}\right)\right] \qquad \left(\begin{matrix}\text{Lemma} \quad \text{A.9} \quad \text{and} \\ N_{t-1} < tK\end{matrix}\right) \tag{90}$$

$$= \sum_{t \in [T]} \mathbb{E}\left[KL\frac{dC_1}{\tau_t}\left(1 + \sqrt{tK}\varsigma^{-1}\sqrt{\|\mathbf{L}^{-1}(\mathbf{y} - \boldsymbol{\mu})\|_2^2}\right)\right] \qquad \left(\begin{matrix}\mathbf{L}^{-1}, \mathbf{y}, \boldsymbol{\mu} \text{ indepen-} \\ \text{dent of } a\end{matrix}\right) \tag{91}$$

$$= \sum_{t \in [T]} KL\frac{dC_1}{\tau_t}\left(1 + \sqrt{tK}\varsigma^{-1}\mathbb{E}\left[\sqrt{\|\mathbf{L}^{-1}(\mathbf{y} - \boldsymbol{\mu})\|_2^2}\right]\right) \tag{92}$$

$$\leq \sum_{t \in [T]} KL\frac{dC_1}{\tau_t}\left(1 + \sqrt{tK}\varsigma^{-1}\sqrt{\mathbb{E}\left[\|\mathbf{L}^{-1}(\mathbf{y} - \boldsymbol{\mu})\|_2^2\right]}\right) \qquad \left(\begin{matrix}\text{Concave Jensen's} \\ \text{inequality}\end{matrix}\right) \tag{93}$$

$$= \sum_{t \in [T]} KL\frac{dC_1}{\tau_t}\left(1 + tK\varsigma^{-1}\right) \qquad \left(\begin{matrix}\|\mathbf{L}^{-1}(\mathbf{y} - \boldsymbol{\mu})\|_2^2 \sim \\ \chi^2 \text{ with at most } (t- \\ 1)K \text{ d.o.f.}\end{matrix}\right) \tag{94}$$

$$\leq \sum_{t \in [T]} \frac{1}{2t^2} \qquad (\text{Assumption on } \tau_t) \tag{95}$$

$$\leq \frac{\pi^2}{12}. \qquad \left(\sum_{t=1}^{\infty} \frac{1}{t^2} = \frac{\pi^2}{6}\right) \tag{96}$$

See the proof of Lemma A.9 for the motivation that $\|\mathbf{L}^{-1}(\mathbf{y} - \boldsymbol{\mu})\|_2^2 \sim \chi^2$. □

Similar to Lemma A.10, we establish a constant bound for the discretization error of the posterior standard deviation:

**Lemma A.11.** *If Assumption 3.3 holds; $\tau_t$ and $\beta_t$ satisfy Eqs. (6b) and (6c) in Assumption 3.4 then, for any sequence of super arms $\mathbf{a}_t \in \mathcal{S}_t$ $t \geq 1$, the posterior standard deviation $\sigma_{t-1}(\mathbf{a})$ satisfies*

$$\sum_{t \in [T]} \mathbb{E}\left[\sqrt{\beta_t}\left(\sigma_{t-1}([\mathbf{a}_t]_{\mathcal{D}_t}) - \sigma_{t-1}(\mathbf{a}_t)\right)\right] \leq \frac{\pi^2}{12}. \tag{97}$$

*Proof.* Note that Eqs. (6b) and (6c) are equivalent to

$$\beta_t K^2 L\frac{dC_1}{\tau_t} \leq \frac{1}{8t^4} \text{ and } \beta_t tK^3 L^2 \frac{d^2C_1^2}{\tau_t^2}\varsigma^{-2} \leq \frac{1}{8t^4}. \tag{98}$$

Then,

$$\sum_{t \in [T]} \mathbb{E}\left[ \sqrt{\beta_t} \left( \sigma_{t-1}([\mathbf{a}]_{\mathcal{D}_t}) - \sigma_{t-1}(\mathbf{a}) \right) \right] \tag{99}$$

$$= \sum_{t \in [T]} \mathbb{E}\left[ \sum_{a \in \mathbf{a}} \sqrt{\beta_t} \left( \sigma_{t-1}([a]_{\mathcal{D}_t}) - \sigma_{t-1}(a) \right) \right] \tag{100}$$

$$\leq \sum_{t \in [T]} \mathbb{E}\left[ \sum_{a \in \mathbf{a}} \sqrt{\beta_t} \sqrt{L \frac{dC_1}{\tau_t} + tKL^2 \frac{d^2 C_1^2}{\tau_t^2} \varsigma^{-2}} \right] \qquad \text{(Lemma A.9)} \tag{101}$$

$$\leq \sum_{t \in [T]} K \sqrt{\beta_t} \sqrt{L \frac{dC_1}{\tau_t} + tKL^2 \frac{d^2 C_1^2}{\tau_t^2} \varsigma^{-2}} \qquad (|\mathbf{a}| \leq K) \tag{102}$$

$$= \sum_{t \in [T]} \sqrt{\beta_t K^2 L \frac{dC_1}{\tau_t} + \beta_t t K^3 L^2 \frac{d^2 C_1^2}{\tau_t^2} \varsigma^{-2}} \tag{103}$$

$$\leq \sum_{t \in [T]} \sqrt{\frac{1}{8t^4} + \frac{1}{8t^4}} \qquad \text{(Eq. (98))} \tag{104}$$

$$\leq \sum_{t \in [T]} \frac{1}{2t^2} \leq \frac{\pi^2}{12}. \qquad \left( \sum_{t=1}^{\infty} \frac{1}{t^2} = \frac{\pi^2}{6} \right) \tag{105}$$

$\square$

**Lemma 3.5.** *If $U_t(\mathbf{a}) = \mu_{t-1}(\mathbf{a}) + \sqrt{\beta_t} \sigma_{t-1}(\mathbf{a})$, Assumption 3.3 holds and $\tau_t$ and $\beta_t$ satisfy Eqs.* (6a) *to* (6c) *in Assumption 3.4, then for any sequence of super arms $\mathbf{a}_t \in \mathcal{S}_t$ $t \geq 1$:*

$$\sum_{t \in [T]} \mathbb{E}\left[ U_t([\mathbf{a}_t]_{\mathcal{D}_t}) - U_t(\mathbf{a}_t) \right] \leq \frac{\pi^2}{6}. \tag{7}$$

*Proof.* Follows by combining Lemmas A.10 and A.11. $\square$

Finally, we are ready to prove the regret bounds for the infinite case:

**Theorem 3.6** (Infinite regret bounds). *If Assumption 3.3 holds and $\tau_t$ satisfies Assumption 3.4, then the Bayesian regret of*

(i) *GP-UCB with $\beta_t = 2\log(\tau_t^d t^2 / \sqrt{2\pi})$ is bounded as $BR(T) \leq \frac{\pi^2}{2} + \sqrt{C_K T K \beta_T \gamma_{TK}}$.*

(ii) *GP-BUCB with $\beta_t = 2\left(\mathrm{erf}^{-1}(1 - 2\eta_t)\right)^2$ for $\eta_t = (2\pi)^{\omega/2} / \left(2\tau_t^{d\omega} t^\xi\right)$, $\xi > \omega > 1$ is bounded as $BR(T) \leq \frac{\pi^2}{3} + \sqrt{C_K T K \beta_T \gamma_{TK}} + C_\omega \cdot \frac{\xi}{\xi - \omega}$ where $C_\omega = \left(\sqrt{\pi}\omega / \sqrt{2e(\omega - 1)}\right)^{1/\omega}$.*

(iii) *GP-TS is bounded as $BR(T) \leq \frac{2\pi^2}{3} + 2\sqrt{C_K T K \beta_T \gamma_{TK}}$.*

*Proof.* (i) Similar to Takeno et al. (2023); Srinivas et al. (2012), we use a fixed discretization $\mathcal{D}_t \subset \mathcal{A}$ for $t \geq 1$. Let $\mathcal{D}_t \subset \mathcal{A}$ be a finite set with $|\mathcal{D}_t| = \tau_t^d$ and each dimension equally divided into $\tau_t$ points with $\tau_t$ satisfying Assumption 3.3. Let $[a]_{\mathcal{D}_t}$ denote the nearest point in $\mathcal{D}_t$ for $a \in \mathcal{A}$ and similarly let $[\mathbf{a}]_{\mathcal{D}_t} = \{[a]_{\mathcal{D}_t} | a \in \mathbf{a}\}$ for $\mathbf{a} \subset \mathcal{A}$.

As Takeno et al. (2023), we decompose the Bayesian regret into several parts:

$$\text{BR}(T) = \sum_{t \in [T]} \mathbb{E} \big[ \underbrace{f(\mathbf{a}_t^*) - f([\mathbf{a}_t^*]_{\mathcal{D}_t})}_{(1)} + \underbrace{f([\mathbf{a}_t^*]_{\mathcal{D}_t}) - U_t([\mathbf{a}_t^*]_{\mathcal{D}_t})}_{(2)} \tag{106}$$

$$+ \underbrace{U_t([\mathbf{a}_t^*]_{\mathcal{D}_t}) - U_t(\mathbf{a}_t^*)}_{(3)} + \underbrace{U_t(\mathbf{a}_t^*) - U_t(\mathbf{a}_t)}_{(4)} \tag{107}$$

$$+ \underbrace{U_t(\mathbf{a}_t) - f(\mathbf{a}_t)}_{(5)} \big] \tag{108}$$

Term (1) can be bounded using Lemma A.8: $\sum_{t \in [T]} \mathbb{E}[f(\mathbf{a}_t^*) - f([\mathbf{a}_t^*]_{\mathcal{D}_t})] \leq \frac{\pi^2}{6}$. Terms (2) and (5) can be bounded using the finite case with $\beta_t = 2\log(|\mathcal{D}_t|t^2/\sqrt{2\pi})$. Then, by Lemmas A.2 to A.4

$$\sum_{t \in [T]} \mathbb{E}[f([\mathbf{a}_t^*]_{\mathcal{D}_t}) - U_t([\mathbf{a}_t^*]_{\mathcal{D}_t}) + U_t(\mathbf{a}_t) - f(\mathbf{a}_t)] \leq \frac{\pi^2}{6} + \sqrt{2(\lambda_K^* + \varsigma^2)TK\beta_T\gamma_{TK}}. \tag{109}$$

Takeno et al. (2023) consider the term $U_t([\mathbf{a}_t^*]_{\mathcal{D}_t}) - U_t(\mathbf{a}_t^*)$ and argue that it is non-positive since $\mathbf{a}_t = \arg\max_{\mathbf{a} \in \mathcal{S}_t} U_t(\mathbf{a})$. Unlike Takeno et al., we do not assume that all arms are available at time $t$ and thus $[\mathbf{a}_t^*]_{\mathcal{D}_t} \in \mathcal{S}_t$ does not necessarily hold. By further decomposing this term into (3) and (4), the same argument can be applied to term (4): $U_t(\mathbf{a}_t^*) - U_t(\mathbf{a}_t) \leq 0$. Then, term (3) can be bounded using Lemma 3.5: $\sum_{t \in [T]} \mathbb{E}[U_t([\mathbf{a}_t^*]_{\mathcal{D}_t}) - U_t(\mathbf{a}_t^*)] \leq \pi^2/6$.

Finally, by combining the bounds for all terms we get that

$$\text{BR}(T) \leq \frac{\pi^2}{2} + \sqrt{C_K T K \beta_T \gamma_{TK}}. \tag{110}$$

(ii) The proof for GP-BUCB is shown by following the steps of GP-UCB and using the finite case for Bayes-GP-UCB (Theorem 3.2 (ii)).

(iii) As in the proof for GP-UCB, assume that we have a discretization $\mathcal{D}_t$ and decompose the Bayesian regret into 4 terms:

$$\text{BR}(T) = \sum_{t \in [T]} \mathbb{E} \big[ \underbrace{f(\mathbf{a}_t^*) - f([\mathbf{a}_t^*]_{\mathcal{D}_t})}_{(1)} + \underbrace{f([\mathbf{a}_t^*]_{\mathcal{D}_t}) - U_t([\mathbf{a}_t^*]_{\mathcal{D}_t})}_{(2)} \tag{111}$$

$$+ \underbrace{U_t([\mathbf{a}_t^*]_{\mathcal{D}_t}) - U_t(\mathbf{a}_t)}_{(3)} + \underbrace{U_t(\mathbf{a}_t) - f(\mathbf{a}_t)}_{(4)} \big]. \tag{112}$$

As in the proof for GP-UCB, term (1) is dealt with using Lemma A.8 and term (2) and (4) are handled as in the finite case (Theorem 3.2 (iii)):

$$\sum_{t \in [T]} \mathbb{E}[(1) + (2) + (3)] \leq \frac{\pi^2}{6} + \frac{\pi^2}{3} + 2\sqrt{C_K T K \beta_T \gamma_{TK}}. \tag{113}$$

To bound term (3), we start by utilizing that $\mathbf{a}_t^*|H_t \overset{d}{=} \mathbf{a}_t|H_t$ and $U_t([\cdot]_{\mathcal{D}_t})|H_t$ is deterministic and thus:

$$\sum_{t \in [T]} \mathbb{E}[(3)] = \sum_{t \in [T]} \mathbb{E}_{H_t} \left[ \mathbb{E}_t \left[ U_t([\mathbf{a}_t^*]_{\mathcal{D}_t}) - U_t(\mathbf{a}_t)|H_t \right] \right] \tag{114}$$

$$= \sum_{t \in [T]} \mathbb{E}_{H_t} \left[ \mathbb{E}_t \left[ U_t([\mathbf{a}_t]_{\mathcal{D}_t}) - U_t(\mathbf{a}_t)|H_t \right] \right] \tag{115}$$

$$\leq \frac{\pi^2}{6} \qquad\qquad\qquad \text{(Lemma 3.5)} \tag{116}$$

Put together, we have that

$$\text{BR}(T) \leq \frac{2\pi^2}{3} + 2\sqrt{C_K T K \beta_T \gamma_{TK}}. \tag{117}$$

$\square$

A.3 ADDITIONAL LEMMAS

**Lemma A.12.** *For any sequence of superarms* $\mathbf{a}_1, \ldots, \mathbf{a}_T$,

$$\sum_{t=1}^{T} \sigma_{t-1}^2(\mathbf{a}_t) \leq 2(\lambda_K^* + \varsigma^2)\gamma_{TK}. \tag{118}$$

*where* $\lambda_K^*$ *is the largest eigenvalue of all possible posterior covariance matrices of size at most* $K$.

*Proof.* This proof follows the proof of Lemma 3 of (Nika et al., 2022). Let $K_t = |\mathbf{a}_t|$ denote the number of base arms selected at time $t$. Similarly, let $N_T = \sum_{t \in [T]} K_t$ denote the number of base arms selected *up to* time $T$. Note that the information gain can be decomposed into two entropy terms: $I(\mathbf{r}_{[T]}; f) = H(\mathbf{r}_{[T]}) - H(\mathbf{r}_{[T]}|f)$.

Since $\mathbf{r}_{[T]}|\mathbf{f}_{[T]} \sim \mathcal{N}(\mathbf{f}_{[T]}, \varsigma^2 I_{K_t})$, $H(\mathbf{r}_{[T]}|\mathbf{f}_{[T]}) = \frac{1}{2}\log|2\pi e\varsigma^2 I_{N_T}|$. The first term can be analyzed by using the chain rule of entropy on the superarms:

$$H(\mathbf{r}_{[T]}) = H(\mathbf{r}_T|\mathbf{r}_{[T-1]}) + H(\mathbf{r}_{[T-1]}) \tag{119}$$

$$= \sum_{t=1}^{T} H(\mathbf{r}_t|\mathbf{r}_{[t-1]}). \tag{120}$$

Then, $\mathbf{r}_t|\mathbf{r}_{[t-1]} \sim \mathcal{N}(\boldsymbol{\mu}_{t-1}, \boldsymbol{\Sigma}_{t-1} + \varsigma^2 I_{K_t})$ where $\boldsymbol{\mu}_{t-1} = [\mu_{t-1}(a)]_{a \in \mathbf{a}_t}$ is the posterior mean vector and $\boldsymbol{\Sigma}_{t-1} = (k_{t-1}(a, a'))_{a,a' \in \mathbf{a}_t \times \mathbf{a}_t}$ is the posterior covariance matrix for superarm $\mathbf{a}_t$ after observing $(\mathbf{a}_1, \mathbf{r}_1), \ldots, (\mathbf{a}_{t-1}, \mathbf{r}_{t-1})$. Let $\lambda_{t,k}$ denote the smallest $k$th eigenvalue of $\boldsymbol{\Sigma}_{t-1}$. Then,

$$H(\mathbf{r}_t|\mathbf{r}_{[t-1]}) = \frac{1}{2}\log\left|2\pi e(\boldsymbol{\Sigma}_{t-1} + \varsigma^2 I_{K_t})\right| \tag{121}$$

$$= \frac{1}{2}\log\left|2\pi e\varsigma^2(\varsigma^{-2}\boldsymbol{\Sigma}_{t-1} + I_{K_t})\right| \tag{122}$$

$$= \frac{1}{2}\log\left|2\pi e\varsigma^2 I_{K_t}\right| + \frac{1}{2}\log\left|\varsigma^{-2}\boldsymbol{\Sigma}_{t-1} + I_{K_t}\right|. \tag{123}$$

Let $\lambda_{t,k}$ denote the smallest $k$th eigenvalue of $\boldsymbol{\Sigma}_{t-1}$. Let $\mathcal{M} = \{\boldsymbol{\Sigma}_{t-1}|\forall t \in [T], \forall \mathbf{a}_1, \ldots, \mathbf{a}_t \in \mathcal{S}\}$ be the set of all possible posterior covariance matrices and let $\lambda_K^* = \sup_{\boldsymbol{\Sigma} \in \mathcal{M}} \max \mathrm{eig}(\boldsymbol{\Sigma})$ be the largest eigenvalue of all eigenvalues of the matrices in $\mathcal{M}$. Recall that $|A + I_n| = \prod_{k \leq n}(\lambda_k + 1)$ for any real and symmetric matrix $A \in \mathbb{R}^{n \times n}$ with eigenvalues $\lambda_1, \ldots, \lambda_n$. Then,

$$\frac{1}{2}\log\left|\varsigma^{-2}\boldsymbol{\Sigma}_{t-1} + I_{K_t}\right| \tag{124}$$

$$= \frac{1}{2}\log\left(\prod_{k=1}^{K_t}\left(\varsigma^{-2}\lambda_{t,k} + 1\right)\right) \tag{125}$$

$$= \frac{1}{2}\sum_{k=1}^{K_t}\log\left(\varsigma^{-2}\lambda_{t,k} + 1\right) \tag{126}$$

$$\geq \frac{1}{2}\sum_{k=1}^{K_t}\frac{\varsigma^{-2}\lambda_{t,k}}{\varsigma^{-2}\lambda_{t,k} + 1} \qquad (\log(x+1) \geq x/(x+1), \forall x > 1) \tag{127}$$

$$\geq \frac{\varsigma^{-2}}{2(\varsigma^{-2}\lambda^* + 1)}\sum_{k=1}^{K_t}\lambda_{t,k} \tag{128}$$

$$= \frac{\varsigma^{-2}}{2(\varsigma^{-2}\lambda^* + 1)}\sum_{a \in \mathbf{a}_t}\sigma_{t-1}^2(a). \qquad \left(\mathrm{Tr}(A) = \sum_{\lambda \in \mathrm{eig}(A)}\lambda\right) \tag{129}$$

Put together, we get that $\sum_{t=1}^{T}\sigma_{t-1}^2(\mathbf{a}_t) \leq 2(\lambda^* + \varsigma^2)I(\mathbf{r}_{[T]}; f)$. Since the maximum information $\gamma_T$ is increasing w.r.t. $T$ and $|\mathbf{a}_t| \leq K$, we get that $\sum_{t=1}^{T}\sigma_{t-1}^2(\mathbf{a}_t) \leq 2(\lambda^* + \varsigma^2)\gamma_{TK}$. $\qquad\square$

**Lemma A.13.** *The inverse error function is lower bounded by*

$$\operatorname{erf}^{-1}(u) \geq \sqrt{-\omega^{-1} \log\left(\frac{1-u}{\vartheta}\right)} \tag{130}$$

*for $u \in [0, 1)$, $\omega > 1$ and $0 < \vartheta \leq \sqrt{\frac{2e}{\pi}} \frac{\sqrt{\omega-1}}{\omega}$.*

*Proof.* According to Theorem 2 of Chang et al. (2011), $\operatorname{erfc}(u) \geq \vartheta \exp(-\omega u^2)$ for $\omega > 1$ and $0 < \vartheta \leq \sqrt{\frac{2e}{\pi}} \frac{\sqrt{\omega-1}}{\omega}$. Since $\operatorname{erf}(u) = 1 - \operatorname{erfc}(u)$, it follows that $\operatorname{erf}(u) \leq 1 - \vartheta \exp(-\omega u^2) =: h(u)$.

In general, if $f(x) \leq g(x)$ then $f^{-1}(x) \geq g^{-1}(x)$. Thus, $\operatorname{erf}^{-1}(u) \geq h^{-1}(u) = \sqrt{-\omega^{-1} \log((1-u)/\vartheta)}$. $\qquad\square$

**Lemma A.14.** *The inverse error function is upper bounded by*

$$\operatorname{erf}^{-1}(u) \leq \sqrt{-\omega^{-1} \log\left(\frac{1-u}{\vartheta}\right)} \tag{131}$$

*for $u \in [0, 1)$, $\vartheta \geq 1$ and $0 < \omega \leq 1$.*

*Proof.* The same arguments as in Lemma A.13 but using Theorem 1 of Chang et al. (2011). $\qquad\square$

# B  ADDITIONAL EXPERIMENTAL DETAILS

## B.1  KERNEL DETAILS

Here, we provide further details on the graph kernel used in the experiments. The original graph Matérn GP of Borovitskiy et al. (2021) defines a GP on the vertices of a weighted and undirected graph. We extend the graph Matérn GP from Borovitskiy et al. (2021) to the edges of a directed graph by considering the incidence graph Laplacian of the line graph $\mathcal{L}(\mathcal{G})$.

Let $\mathbf{W}_{\mathcal{L}} \in \mathbb{R}^{|\mathcal{E}| \times |\mathcal{C}|}$ denote the weight matrix of $\mathcal{L}(\mathcal{G}) = (\mathcal{E}, \mathcal{C})$ where $\mathcal{E}$ is the set of edges and $\mathcal{C}$ is the set of all connections in the network. The weight $W_{\mathcal{L}, e_1, e_2}$ is set to $\bar{\ell}/\ell_{e_1}$ where $\bar{\ell}$ is the average length of all edges and $\ell_{e_1}$ is the length of edge $e_1$. We replace the ordinary graph Laplacian used by Borovitskiy et al. (2021) with the incidence Laplacian:

$$\boldsymbol{\Delta}_I = \mathbf{B}\mathbf{B}^{\top}, \tag{132}$$

where the incidence matrix $\mathbf{B} \in \mathbb{R}^{|\mathcal{E}| \times |\mathcal{C}|}$ has entries

$$B_{e,c} = \begin{cases} -W_{\mathcal{L}, e_1, e_2} & \text{if } e = e_1, \\ W_{\mathcal{L}, e_1, e_2} & \text{if } e = e_2, \quad \forall e \in \mathcal{E}, c = (e_1, e_2) \in \mathcal{C}. \\ 0 & \text{otherwise} \end{cases} \tag{133}$$

Let $\boldsymbol{\Delta}_I = \mathbf{U}_I \boldsymbol{\Lambda}_I \mathbf{U}_I^{\top}$ denote the eigendecomposition of $\boldsymbol{\Delta}_I$, then the graph Matérn GP of the edges is given by

$$\mathbf{f} \sim \mathcal{N}\left(0, \mathbf{U}_I \left(\frac{2\nu_G}{\kappa_G^2}\mathbf{I} + \boldsymbol{\Lambda}_I\right)^{-\nu} \mathbf{U}_I^{\top}\right). \tag{134}$$

Recall that $k_f : \mathbb{R}^d \times \mathbb{R}^d \to \mathbb{R}$ denotes a feature kernel which measures the similarity between the contexts of the edges. The feature kernel is an ordinary Matérn kernel with fixed $\nu = 5/2$ but tunable outputscale $\sigma_f$ and lengthscales $\boldsymbol{\ell}_f \in \mathbb{R}_+^d$ for each dimension:

$$k_f(x_e, x_{e'}) := \sigma_f \frac{2^{1-\nu}}{\Gamma(\nu)} \left(\sqrt{2\nu}D\right)^{\nu} K_{\nu}\left(\sqrt{2\nu}D\right), \tag{135}$$

Table 2: Vehicle and environmental parameters for the energy model.

| Variable | Value | Unit |
|---|---|---|
| Mass $m$ | 1830 | kg |
| Rolling resistance coefficient $C_r$ | 0.01 | |
| Front surface area $A$ | 2.6 | m$^2$ |
| Air drag coefficient $C_d$ | 0.35 | |
| Power train efficiency $\eta^+$ | 0.98 | |
| Recuperation efficiency $\eta^-$ | 0.96 | |
| Gravitational acceleration $g$ | 9.82 | m/s$^2$ |
| Air density $\rho$ | 1.2 | kg/m$^3$ |

where $x_e$ denotes the feature vector of edge $e$ and the feature distance $D$ between edge $e$ and $e'$ is given by

$$D = \sqrt{(x_e - x_{e'})^\top \operatorname{diag}(\boldsymbol{\ell}_f)^{-2} (x_e - x_{e'})}. \tag{136}$$

The kernels, the SVGP model and Algorithm 3 was implemented using GPyTorch (Gardner et al., 2018).

## B.2 ROAD NETWORK

The set of available paths was restricted to edges within the largest strongly connected component. This mainly removed road segments in inaccessible areas and does not affect the navigational challenge. The route Luxembourg A starts in edge `-31118#2` and ends in edge `--32646#1`. The route Luxembourg B starts in edge `-30436#5` and ends in edge `-30946#0`. Similarly, the route Monaco A starts in edge `-30558` and ends in edge `-32888#0` whilst Monaco B starts in edge `-32166#0` and ends in edges `--32940#0`. For simplicity, the start and end points are edges since the shortest path was computed using the line graph $\mathcal{L}(\mathcal{G})$.

## B.3 DETAILED PARAMETER VALUES

In this section, we further specify the vehicle, environmental and algorithmic parameters used. We use the default parameters for electric vehicles provided by SUMO (Lopez et al., 2018), see Table 2.

The graph kernel is initialized with parameters $\nu_G = 2$, $\kappa_G = 1$ and $\sigma_G$ set according to the prior. The natural gradient descent learning rate is set to 0.1 whilst the Adam learning rate is set to 0.01. The GP model uses a batch size $B$ of 2500 and 1 gradient step per optimization procedure. The number of inducing points is set to 1000.

## C ADDITIONAL EXPERIMENTAL RESULTS

### C.1 IMPACT OF LENGTHSCALE

In this section, we provide the full results for the lengthscale experiments in Section 4.2. The cumulative regret over time is visualized in Fig. 4 and the final cumulative regret as a function on the lengthscale $\ell$ is visualized in Fig. 5.

### C.2 VISUALIZATION OF EXPLORATION

In this section, we provide visualization of the routes selected by the algorithms. See Figs. 6 to 9 for visualization on Lux. A, Lux B, Mon. A and Mon. B, respectively. According to the results, the TS variants are able to find sophisticated paths with significantly less exploration compared to BUCB and UCB. This observation implies the sample efficiency of TS methods.

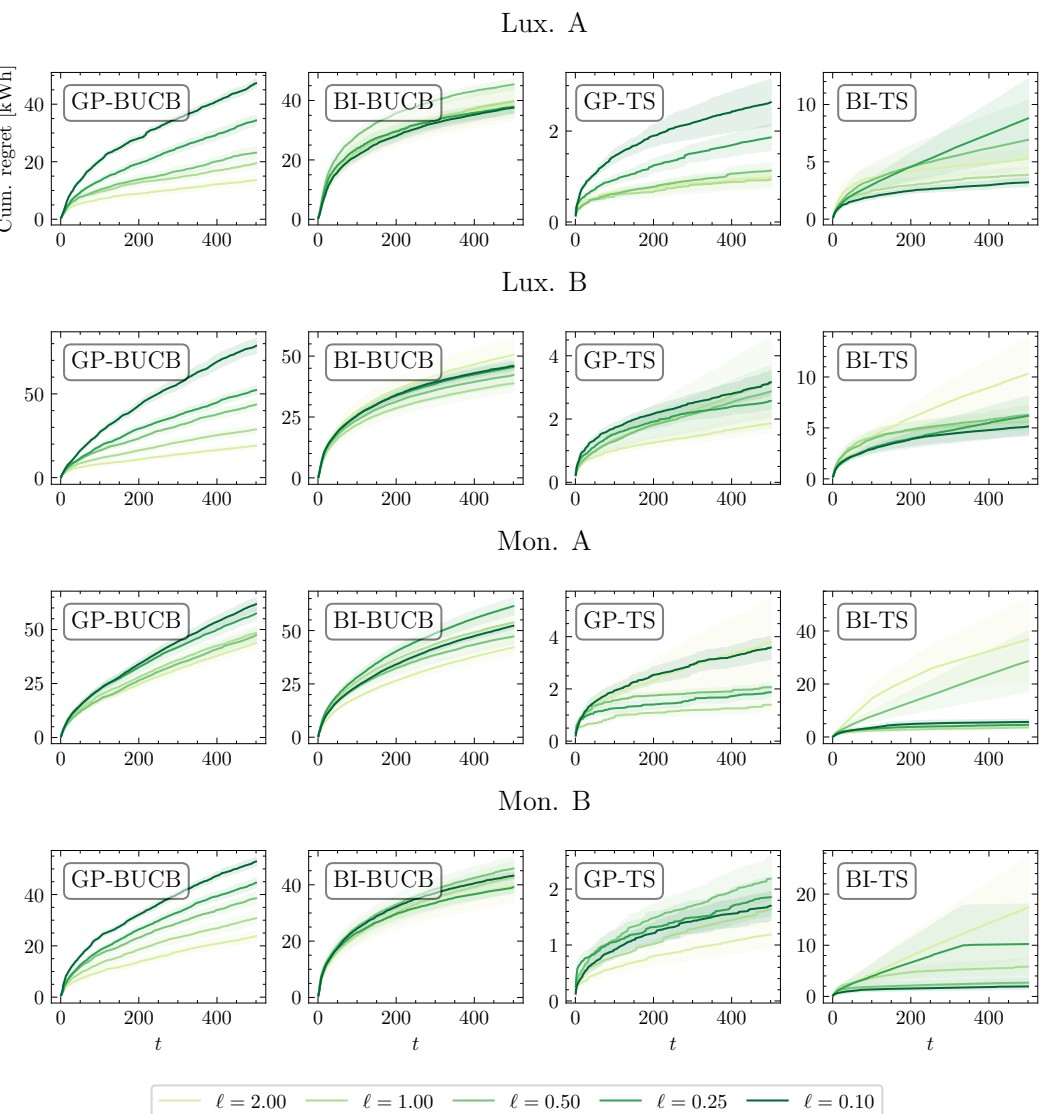

Figure 4: Cumulative regret of GP-BUCB, BI-BUCB, GP-TS and BI-TS for varying prior lengthscale values $\ell$.

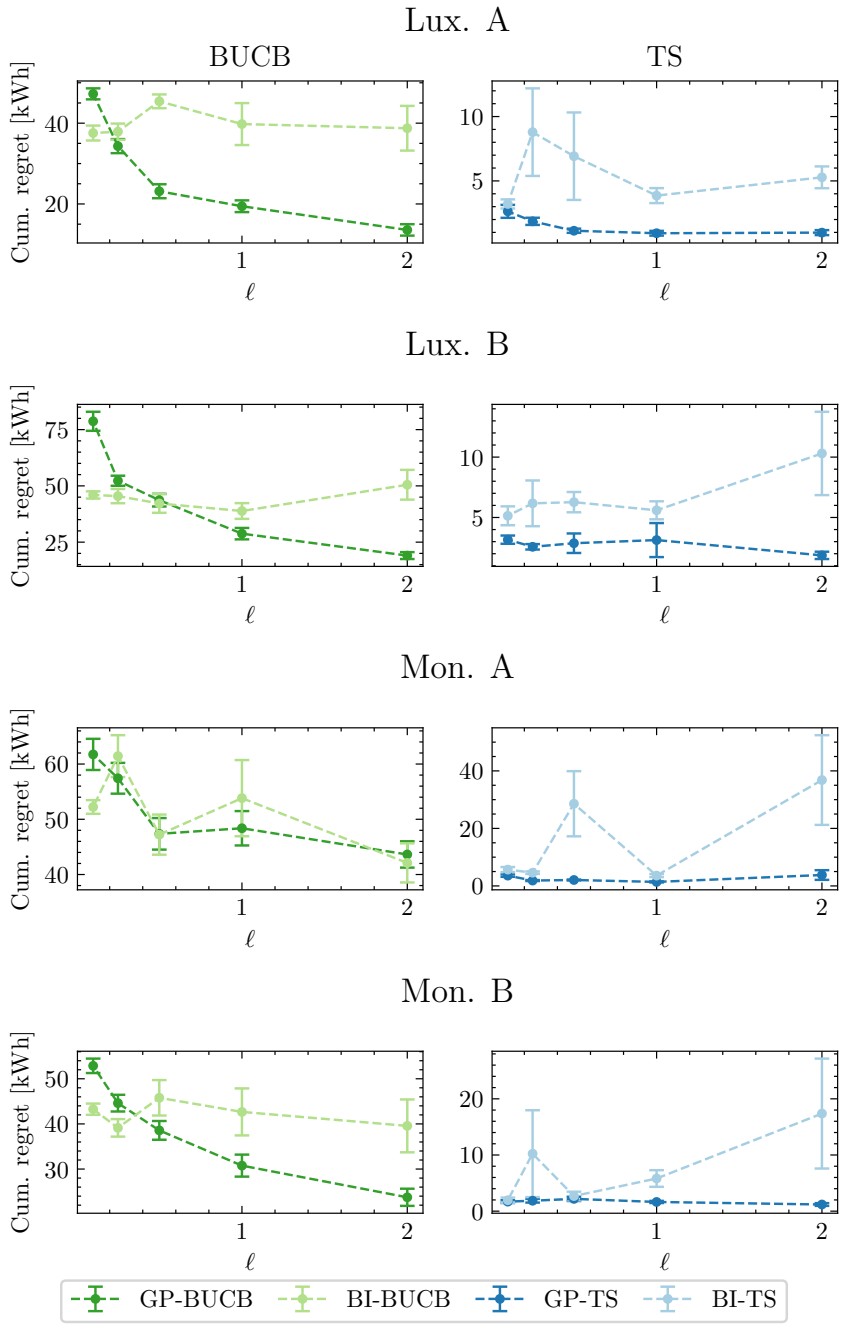

Figure 5: Cumulative regret at $t = 500$ for varying prior lengthscale values. Errorbars correspond to $\pm 1$ standard error.

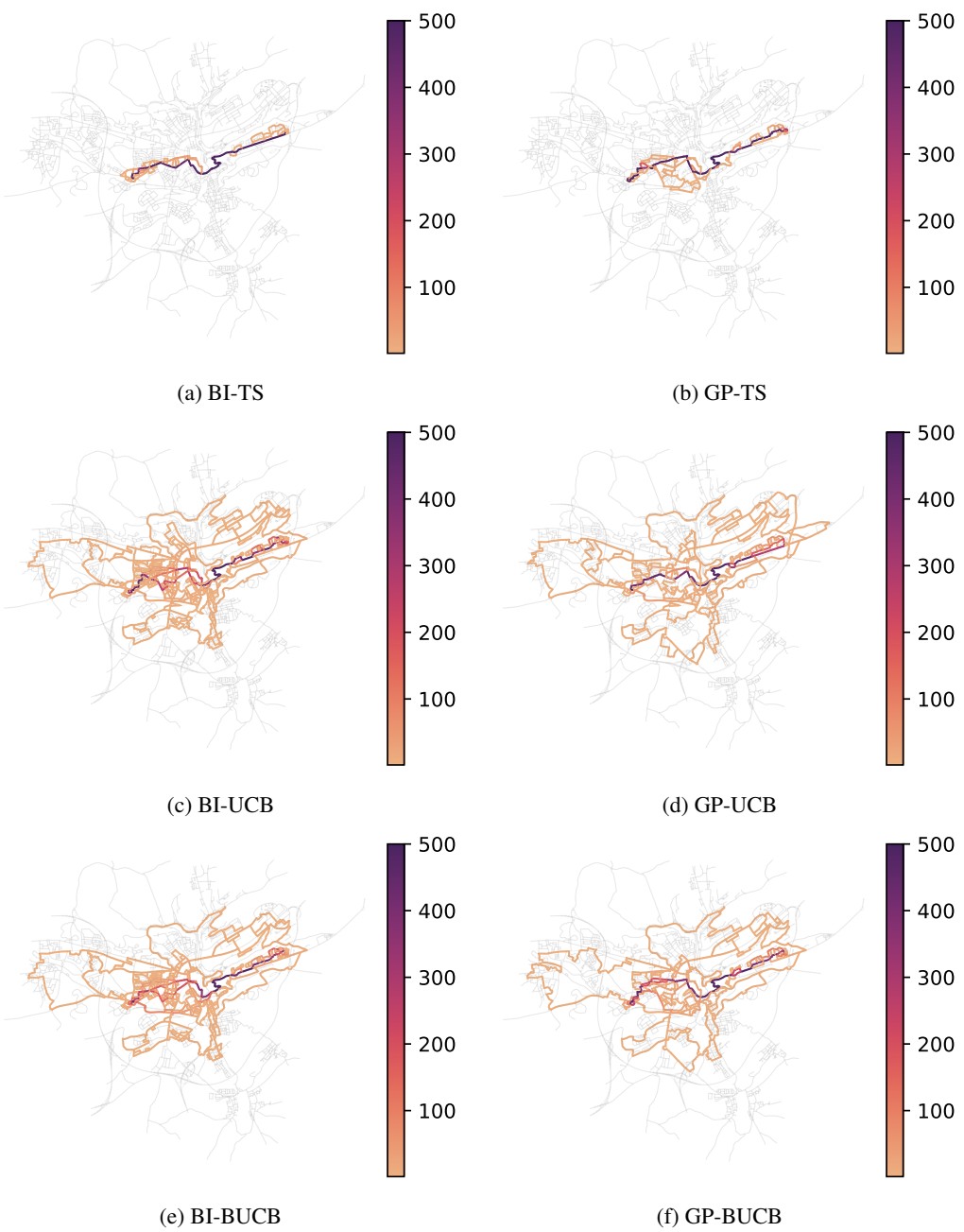

Figure 6: Exploration of Luxembourg A.

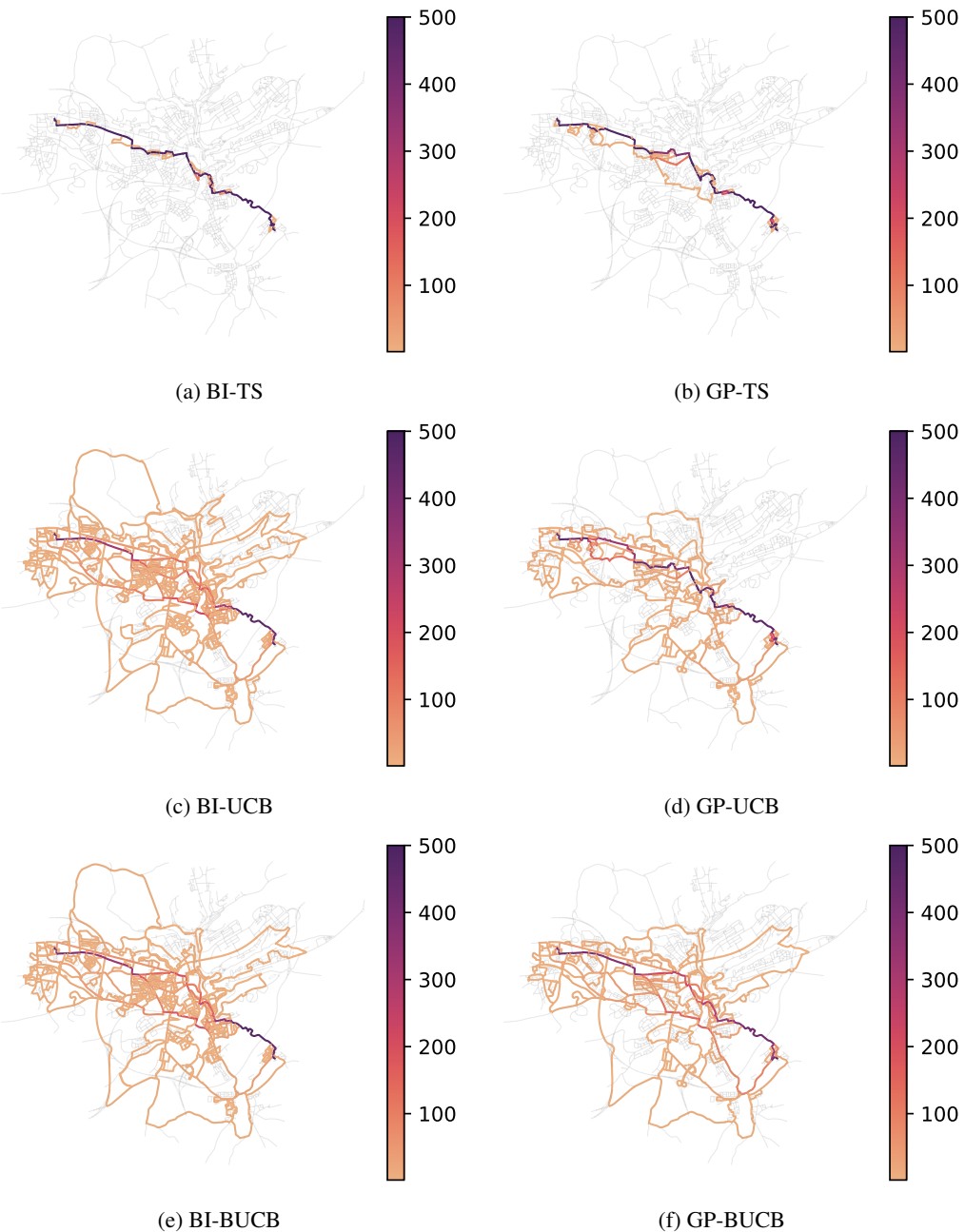

Figure 7: Exploration of Luxembourg B.

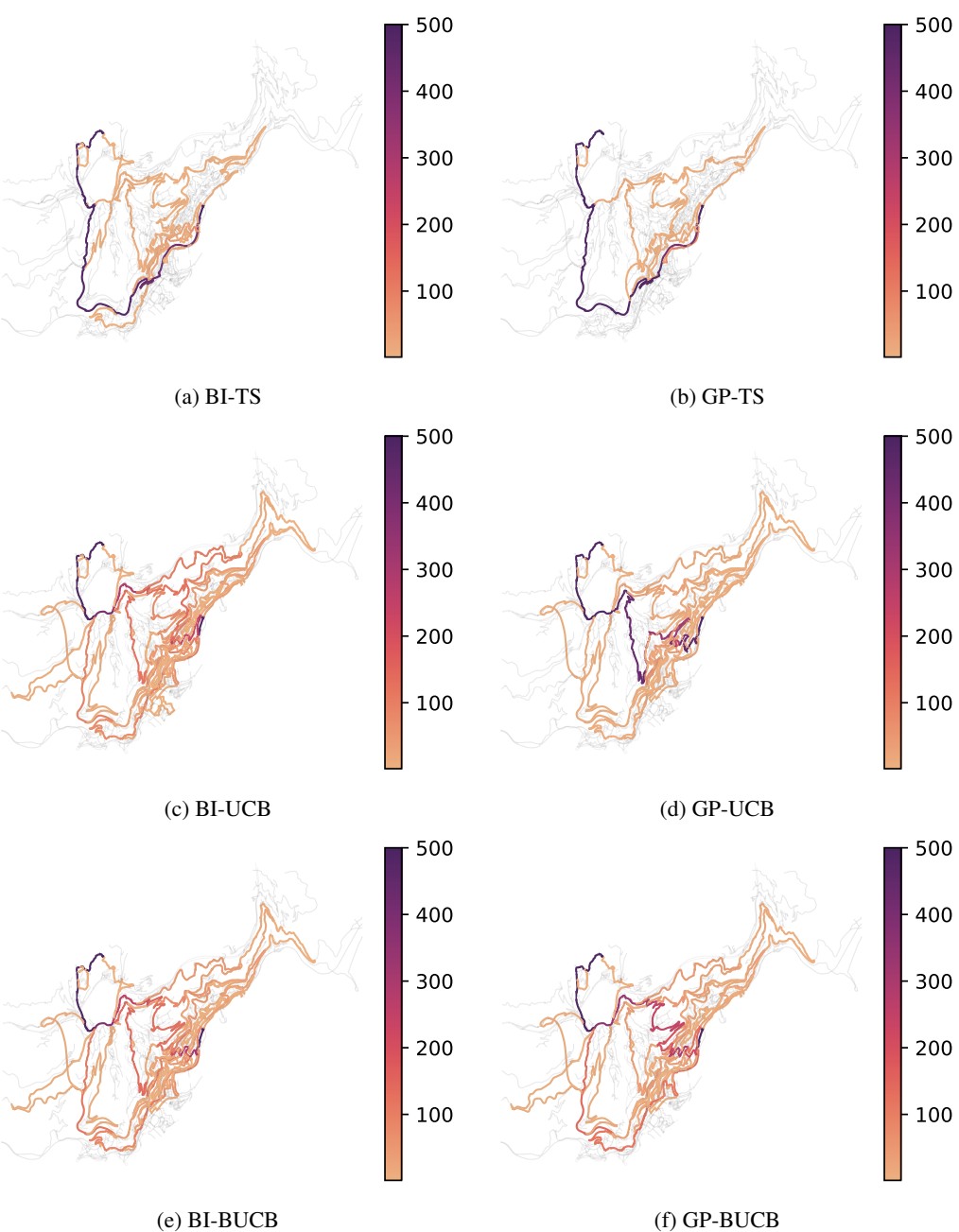

Figure 8: Exploration of Monaco A.

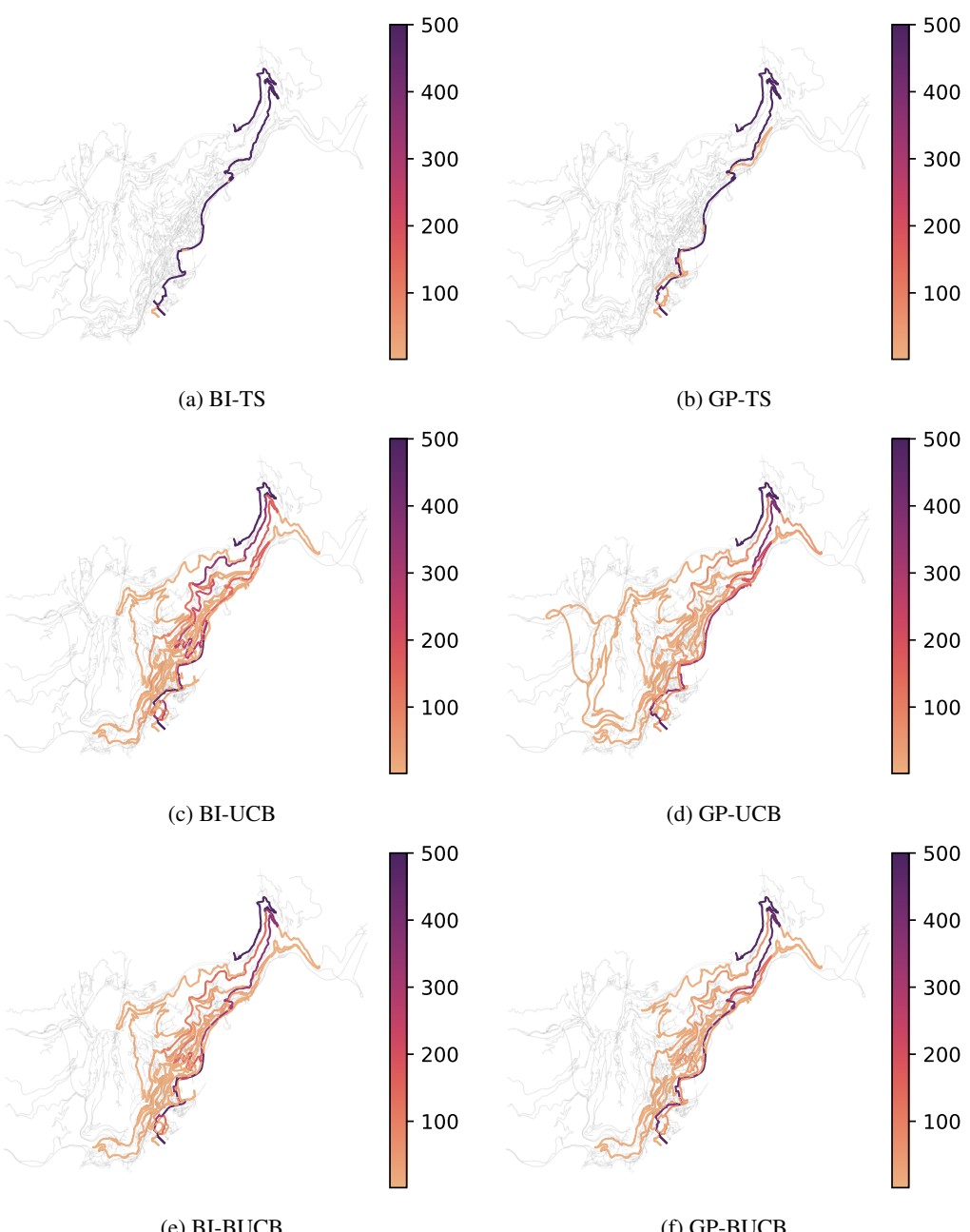

(a) BI-TS

(b) GP-TS

(c) BI-UCB

(d) GP-UCB

(e) BI-BUCB

(f) GP-BUCB

Figure 9: Exploration of Monaco B.

