# OpenReview forum: "Bayesian Analysis of Combinatorial Gaussian Process Bandits"
_ICLR.cc/2025/Conference — ICLR 2025 Poster_

### Official Review · Reviewer_8zLg · 2024-10-29

**Soundness:** 3
**Presentation:** 4
**Contribution:** 2
**Rating:** 5
**Confidence:** 2

**Summary:**

The paper studies Gaussian process bandits in the contextual volatile semi-bandit setting. The contribution of the paper is mainly theoretical as it provides novel Bayesian regret bounds for previously designed algorithms. In addition, there is an interesting application of their framework in online energy efficient navigation. This experimental application builds on top of the previously designed experiment of the same application in bandit papers.

**Strengths:**

The paper is building on top of other previously published frameworks, however, it is not a straightforward extension of the previous works.

The experiments (application of their framework) in online energy-efficient navigation problem seem to have added some novelties and value to the paper.

The paper is written in an excellent way. The explanations for the most important parts of the algorithms are clear. Also the similarities and differences (novelties) of their framework in comparison to the state-of-the-art is clarified properly.

**Weaknesses:**

No synthetic data experiment. Not even in the supplementary material. In my opinion, synthetic data experiments can significantly add to the development of intuitions about the framework. Also since you have much more control over the creation of the data, it can reveal interesting properties of the framework [in comparison with state-of-the-art].

Also, I could not find an experiment with the horizon more than 500 rounds. I am curious about the performance of the frameworks as the horizon goes well beyond T=500. I believe that proper comparison of bandit frameworks [most of the times] comes with running the experiments for long horizons.

I did not notice any discussion in the paper about possible extensions and future directions and further impacts of their research.

**Questions:**

Does the type of directed graph affect the applicability of the framework? For instance, how does the graph [being cyclic or acyclic] affect the performance of the framework?

I did not notice any discussion in the paper about possible extensions and future directions and further impacts of their research, not even in the supplamentary section. Why? Can you please clarify?

---

> ### Author Response · Authors · 2024-11-18
>
> We thank **reviewer 8zLg** for their review and feedback. We are pleased to hear that the reviewer finds value in our experimental setup and greatly appreciates our writing. Below, we address the points raised by the reviewer.
>
> **W1: No synthetic data experiment. Not even in the supplementary material. In my opinion, synthetic data experiments can significantly add to the development of intuitions about the framework. Also since you have much more control over the creation of the data, it can reveal interesting properties of the framework [in comparison with state-of-the-art].**
>
> All our experiments are performed in a controlled setting by sampling the expected reward from a Gaussian process and the noise from i.i.d. zero-mean Gaussians. The mean, kernel and noise parameters are based on realistic values given by the deterministic energy model in Eq. (8). The graph and its corresponding edge features are also based on the real-world road networks of Luxembourg and Monaco. Note that we could have set the parameters and the graph arbitrarily. By using realistic values, the obtained regrets can be interpreted in terms of the energy saved and we do not lose much, if any, control over the data creation. Additionally, this still corresponds to a synthetic data generation process.
>
> In the experiments, we study interesting properties of the algorithms, such as the impact of the BayesUCB and UCB parameter values. We find that GP-BayesUCB has parameters that can be tuned to lower the regret whilst maintaining theoretical guarantees, unlike GP-UCB.  Additionally, we vary the lengthscale parameter in the GP prior and demonstrate the difference between our method and previous work is most pronounced with longer lengthscales since it simplifies the problem. See our answer to reviewer 2DSA for further discussions on the connection between theory and empirical results.
>
> **W2: Also, I could not find an experiment with the horizon more than 500 rounds. I am curious about the performance of the frameworks as the horizon goes well beyond T=500. I believe that proper comparison of bandit frameworks [most of the times] comes with running the experiments for long horizons.**
>
> For any bandit experiments on a limited budget, there is a tradeoff between the number of experiments, samples and horizon length. Note that even with $T=500$, the best-performing method (GP-TS) usually reaches saturation. The main purpose of introducing GP-methods to the energy-efficient navigation problem was to improve the sample efficiency compared to Åkerblom et al. (2023) by taking advantage of correlations. Therefore, we chose to provide more experiments rather than longer experiments to demonstrate that the superior sample efficiency holds across different routes, networks, UCB parameter values and lengthscales.
>
> **W3: I did not notice any discussion in the paper about possible extensions and future directions and further impacts of their research.**
>
> See **Q2**.
>
>
> **Q1: Does the type of directed graph affect the applicability of the framework? For instance, how does the graph [being cyclic or acyclic] affect the performance of the framework?**
>
> In general, the type of directed graph should not affect the applicability of the overall framework and its theoretical analysis. However, given some domain knowledge on the type of graph one may tailor the steps in Algorithm 1 to improve the performance. For example, given an acyclic graph one does not need to guarantee that the weights are positive and one could therefore replace Dijkstra's algorithm with the Bellman-Ford algorithm for small enough graphs.
>
> **Q2: I did not notice any discussion in the paper about possible extensions and future directions and further impacts of their research, not even in the supplamentary section. Why? Can you please clarify?**
>
> An important future direction of this research is to study the multi-agent setting where the vehicle can learn from observations given by a fleet of vehicles. The methods we introduce should be able to learn from fewer vehicles since by taking correlations into account, we do not need to explore the entire network. Our motivation for considering the online energy-efficient navigation problem is to extend the effective range of electric vehicles by identifying efficient routes quickly. We hope that our work can lead to reduced range anxiety among consumers and speed up the transition to a sustainable transportation system.
>
> We will try to add an additional section to the supplementary material before the discussion phase ends.

---

### Official Review · Reviewer_WnA6 · 2024-11-04

**Soundness:** 2
**Presentation:** 2
**Contribution:** 1
**Rating:** 5
**Confidence:** 4

**Summary:**

The paper claims to present a novel Bayesian regret bounds for GP-UCB and GP-TS in combinatorial, volatile and infinite arms setting. Further they present the experimental results for a real world application of online energy efficient navigation.

**Strengths:**

The paper provides the bounds for the Bayesian regret for GP-BUCB, GP-UCB and GP-TS.

**Weaknesses:**

1. Though the work claims to present the bounds for volatile case but the proof for the bounds do not seem to consider it. As an example what would happen when the best arm is not present among the observed arms?
2. Not significant contribution, the paper mainly builds on the works of Russo & Roy 2014, Srinivas et al 2012 and Takeno et al 2023, where in to compute the Bayesian regret one only needs to compute the expectation over the high probability regret bounds given by the above works.
3. Lemma 3.1 the results are considered for different regimes of horizons for different cases of the ratio, why not choose the limits as 1 to T for the 3rd case, wouldn't that be a tighter bound?

**Questions:**

See the weakness section

---

> ### Author Response · Authors · 2024-11-18
>
> We thank **reviewer WnA6** for their review and examination of our work. The reviewer is not convinced that we properly consider the volatile case however we assert that **we do consider the volatile case** and we hope the reviewer will engage in further discussions so that any confusion can be clarified.
>
> **W1: Though the work claims to present the bounds for volatile case but the proof for the bounds do not seem to consider it. As an example what would happen when the best arm is not present among the observed arms?**
>
> The volatile case is considered for all the proofs in the paper. Note that the best arm, $\mathbf{a}_t^*$, is defined as the best arm among the available arms (see line 140) and we use the subscript $t$ to indicate that it varies over time. For clarification, could you point to a specific equation, lemma or theorem where the volatile case is not considered?
>
> **W2: Not significant contribution, the paper mainly builds on the works of Russo & Roy 2014, Srinivas et al 2012 and Takeno et al 2023, where in to compute the Bayesian regret one only needs to compute the expectation over the high probability regret bounds given by the above works.**
>
> The Bayesian regret cannot simply be computed by taking the expectation over the high-probability bounds given by Srinivas et al. (2012), Russo \& Roy (2014), and Takeno et al. (2023). In fact, neither Russo \& Roy nor Takeno et al. provide high probability bounds, they both provide Bayesian regret bounds. As we state in the introduction, our contribution is to extend existing Bayesian regret bounds for GP-bandits to new and practically important settings: the infinite, volatile, and combinatorial settings. Additionally, we introduce the first regret bound for GP-BayesUCB (which was introduced by Nuara et al., 2018). Note that previous results only held for finite volatile (Russo \& Roy, 2014) or infinite non-volatile settings (Srinivas et al., 2012, Takeno et al., 2023, 2024, Kandasamy et al., 2018). The closest similar work to ours (Nika et al., 2022) provide frequentist bounds for GP-UCB in the same setting but we establish Bayesian regret bounds in a unified manner for GP-UCB, GP-TS and GP-BUCB.
>
> For the finite case, we adapt the results of Chang et al. (2011) to bound the inverse error function $\text{erf}^{-1}(u)$ (Lemma 3.1) and obtain regret bounds for GP-BUCB with more flexible parameters compared to GP-UCB. We provide a demonstration in our experiments of how the choice of parameters can lower the regret but maintain theoretical guarantees. Our technical contribution for the infinite setting is to provide an analysis of the discretization error $U_t([\mathbf{a}]_{\mathcal{D}_t}) - U_t(\mathbf{a})$ (Lemma 3.5) which is the key step to allow volatile arms.
>
> **W3: Lemma 3.1 the results are considered for different regimes of horizons for different cases of the ratio, why not choose the limits as 1 to T for the 3rd case, wouldn't that be a tighter bound?**
>
> Taking the limits $[1,T]$ instead of $[1,\infty]$ does not yield a meaningfully tighter bound. On page 16, line 846, we would obtain an additional term ($\propto t^{1-\xi/\omega}$) that would go to zero as $T \rightarrow{} \infty$ . Note also that in the equivalent lemma for GP-UCB and GP-TS (Lemma A.2) we also take the limit $T \rightarrow{} \infty$ to obtain a constant bound, see lines 783-786. By doing the same in Lemma 3.1, we are consistent with the other algorithms and previous work.

---

> > ### Comment · Reviewer_WnA6 · 2024-11-27
> > **Feedback**
> >
> > 1. All the results related to the bounds stated in the paper would still hold if the case was non-volatile. It would be better to elaborate on what kind of volatility is being considered by assigning a certain distribution on the set of arms being observed or not observed at any time t. I think such a consideration would have an significant impact on the results and would be tuned more towards the volatile arms. Additionally as the question is stated - let's say an optimal arm is not observed during the initial set of would that not account for a constant regret of the true optimal - observed optimal? which is not reflected in the bounds.
> >
> > 2. The paper does not show a significant contribution because the over all reward considered at any time t is sum of the rewards of each individual arm and which would be similar to considering a single arm bandit frame work with the set of observed arms varying after every K iterations and all the results by previous work would just follow. The only novel contribution is due to Lemma A9  and A.12 which brings in the factor of largest eigenvalue of all the possible covariance matrices.
> >
> > 3. The same argument can be applied to the case one where the as T tends to $\infty$ to case one and which would lead to a constant value? Additionally, The LHS of the bound does not say that it is considering the case of asymptotic bound.
> >
> > Thank you for the response to the questions. I will keep my score.

---

> > > ### Author Response · Authors · 2024-11-28
> > > **Reply to reviewer WnA6**
> > >
> > > **W1**
> > > > "All the results related to the bounds stated in the paper would still hold if the case was non-volatile."
> > >
> > > Yes, this is because the case $\mathcal{A}_t=\mathcal{A}$ is a special case of the volatile setting $\mathcal{A}_t \subseteq \mathcal{A}$. In the infinite case, we require the number of discretization points $\tau_t$ to satisfy additional inequalities (Eq (6a-c) in Assumption 3.5) that are not necessary for the non-volatile case. The confidence parameter $\beta_t$ ($\propto \log \tau_t$) is therefore larger in the volatile case than would be necessary in the non-volatile case. Our bounds do consider the volatile case as evident from them containing $\beta_T$.
> > >
> > > > "It would be better to elaborate on what kind of volatility is being considered by assigning a certain distribution on the set of arms being observed or not observed at any time t. I think such a consideration would have an significant impact on the results and would be tuned more towards the volatile arms."
> > >
> > > Our results hold for any adversarial or random selection of the available base arms $\mathcal{A}_t$ and available super arms $\mathcal{S}_t$. Assigning a specific distribution would detract from the generality of our results.
> > >
> > > > "Additionally as the question is stated - let's say an optimal arm is not observed during the initial set of would that not account for a constant regret of the true optimal - observed optimal? which is not reflected in the bounds."
> > >
> > > **This does not need to be reflected in the bounds.** To clarify the terminology, the set of *feasible* and *available* super arms at time $t$ is $\mathcal{S}\_t \subset 2^{\mathcal{A}\_{t}}$ where $\mathcal{A}\_t$ is the *available* base arms at time $t$. Again, we reiterate that the optimal super arm at time $t$ is defined as $\mathbf{a}^*\_t=\arg\max_{\mathbf{a} \in \mathcal{S}\_{t}} \sum_{a \in \mathbf{a}} f(a)$ since the agent cannot select a better arm than $\mathbf{a}_t^*$ at time $t$. **Note that** $\mathbf{a}^*\_{t}$ **varies over time in the volatile arm setting as indexed by $t$**. Hence, the agent can always obtain an instant regret of $0$. Defining the optimal arm as the best available arm is standard in volatile bandits, see Qin et al. (2014), Russo \& Roy (2014), Chen et al. (2018) and Nika et al. (2020; 2022). As a comparison, contextual bandit problems always compare against the best arm *given the current context* - which is equivalent to the best available arm for volatile bandits.
> > >
> > > - We would like to revisit the reviewer's original claim in weakness 1: "... the proof for the bounds do not seem to consider [the volatile case]". We still assert that **we do consider the volatile case in both the proofs and bounds.** After considering the points we raise, **does the reviewer still stand by this claim?**
> > >
> > > **W2**
> > > > "The paper does not show a significant contribution because the over all reward considered at any time t is sum of the rewards of each individual arm and ... The only novel contribution is due to Lemma A9  and A.12 which brings in the factor of largest eigenvalue of all the possible covariance matrices."
> > >
> > > Linear rewards are not uncommon in the literature, see Kveton et al. (2015), Wen et al. (2015), Russo & Roy (2016), and Åkerblom et al. (2023). As for contributions beyond the combinatorial setting, we establish the first regret bound for GP-BayesUCB. In addition, for GP-TS and GP-UCB, we fill the unaddressed gap of infinite and volatile GP-bandits.
> > >
> > > Kveton et al. "Tight Regret Bounds for Stochastic Combinatorial Semi-Bandits." Proceedings of the Eighteenth International Conference on Artificial Intelligence and Statistics, 2015.
> > >
> > > Combes et al. "Combinatorial Bandits Revisited." Advances in Neural Information Processing Systems 28, 2015.
> > >
> > > Wen et al. "Efficient Learning in Large-Scale Combinatorial Semi-Bandits." Proceedings of the 32nd International Conference on Machine Learning, PMLR 37:1113-1122, 2015.
> > >
> > > Russo & Roy. "An Information-Theoretic Analysis of Thompson Sampling." Journal of Machine Learning Research 17, no. 68 (2016).
> > >
> > > > "... which would be similar to considering a single arm bandit frame work with the set of observed arms varying after every K iterations and all the results by previous work would just follow."
> > >
> > > **This hypothetical solution would not work**, as one would need to account for the delayed feedback. Even if one did account for the delayed feedback, there are no previous Bayesian regret bounds considering the infinite and volatile setting.
> > >
> > > **W3**
> > >
> > > **No, this is incorrect.** The same argument cannot be applied to case one where $\xi/\omega < 1$ as $\lim_{T \rightarrow \infty} T^{1 - \xi / \omega} = \infty$. Letting $T \xrightarrow{} \infty$ for bounded right hand sides is standard procedure within the literature. We encourage the reviewer to compare against the proof of Theorem 2 in Srinivas et al. (2012), Lemma 2 in Russo \& Roy (2014), Theorem 11 in Kandasamy et al. (2018), Theorem B.1 (Eq. (5)) in Takeno et al. (2023).

---

### Official Review · Reviewer_2DSA · 2024-11-04

**Soundness:** 3
**Presentation:** 3
**Contribution:** 3
**Rating:** 8
**Confidence:** 3

**Summary:**

This paper investigates the combinatorial volatile Gaussian process (GP) semi-bandit problem and provides the first Bayesian regret bounds for the GP-BayesUCB algorithm. In addition to this novel contribution, the authors extend their theoretical analysis to include Bayesian cumulative regret bounds for the GP-UCB and GP-TS algorithms, effectively addressing a notable research gap as highlighted in Table 1. To demonstrate the practical relevance of their framework, the authors apply their methods to a real-world problem: online energy-efficient navigation.

**Strengths:**

1.	Clear and Structured Presentation:
The paper is well-written, with clear explanations and illustrations of the research gaps. The novelty of this work is effectively communicated, making it accessible even to readers who may not be deeply familiar with the field.
2.	Solid Theoretical Contributions:
The authors provide rigorous theoretical analysis and establish new Bayesian regret bounds for multiple algorithms, including GP-BayesUCB, GP-UCB, and GP-TS. The paper addresses a significant gap in the literature by formalizing regret bounds for these settings. Full proofs are provided in the appendices, showcasing the depth of their analysis (though the correctness of these proofs was not verified).
3.	Practical Application:
The real-world application of their framework to online energy-efficient navigation is both relevant and interesting. It demonstrates the practical utility of their theoretical advancements and highlights the potential for real-world impact.

**Weaknesses:**

1.	Lack of Discussion on Theoretical Challenges:
While the paper provides new theoretical results, it does not clearly articulate the specific challenges encountered in deriving these results for GP-BayesUCB, GP-UCB, and GP-TS. A discussion on the theoretical hurdles and how they were addressed would provide valuable insight into the novelty and difficulty of these contributions.
2.	Reproducibility Concerns:
No code is provided for the experiments. This absence raises concerns about the reproducibility of the empirical results.

**Questions:**

1.	Connection Between Theory and Empirical Results:
The online energy-efficient navigation application is a compelling demonstration of the framework’s practical utility. However, it would be helpful to clarify how the empirical results relate to the theoretical findings. Specifically, can the empirical results be used to verify or illustrate key observations from the theoretical analysis? If this connection is not direct, could you design controlled simulated experiments that more explicitly validate the theoretical regret bounds or insights?
2.	Extended Comparison in Table 1:
Including the regret rates alongside the regret bounds in Table 1 would greatly enhance its utility. This addition would allow readers to quickly compare the performance of different algorithms in terms of their theoretical guarantees. An extended table with this information would provide a clearer overview of the contributions and situate the work more firmly within the existing literature.
3.	Discussion of Theoretical Challenges:
As mentioned in the weaknesses, a dedicated section or paragraph discussing the theoretical challenges faced in deriving the regret bounds for GP-BayesUCB, GP-UCB, and GP-TS would add significant value. This discussion could cover aspects such as handling the volatility in combinatorial settings, managing the complexities introduced by semi-bandit feedback, or other technical hurdles specific to these algorithms.

---

> ### Author Response · Authors · 2024-11-18
>
> We thank **reviewer 2DSA** for their review and feedback. We are happy to hear that the reviewer appreciates our theoretical contributions and its practical application. Below, we address the points raised by the reviewer.
>
> **W1 Lack of Discussion on Theoretical Challenges**
>
> See **Q3**.
>
> **W2 Reproducibility Concerns: No code is provided for the experiments. This absence raises concerns about the reproducibility of the empirical results.**
>
> The code will be made publicly available for the camera-ready version. Our experiments relies only upon openly accessible data and we have provided references on how to access it. In addition, the data processing and algorithms used are detailed in the main text and supplementary material.
>
> **Q1 Connection Between Theory and Empirical Results: The online energy-efficient navigation application is a compelling demonstration of the framework’s practical utility. However, it would be helpful to clarify how the empirical results relate to the theoretical findings. Specifically, can the empirical results be used to verify or illustrate key observations from the theoretical analysis? If this connection is not direct, could you design controlled simulated experiments that more explicitly validate the theoretical regret bounds or insights?**
>
> Yes, the empirical results can be used to verify or illustrate key observations from the theoretical analysis. First, as the theory suggests and the results in Figure 1 validate, the algorithms obtain sublinear regret w.r.t $T$.
>
> Second, our theoretical analysis shows that the GP-BayesUCB provides more flexible parameters than GP-UCB. It is generally accepted that GP-UCB algorithms tends to overexplore due to too large theoretical confidence intervals. In the right column of Figure 3, we show that the confidence parameter $\beta_t$ of GP-BayesUCB is lower than that of GP-UCB, and additionally we can tune it to be even lower. In practice, (left and middle column of Figure 3), we also show experimentally that GP-BayesUCB obtains lower regret than GP-UCB.
>
> Third, as discussed by Russo \& Roy (2014), the performance of UCB algorithms depend on designing tight confidence bounds whereas the regret of Thompson sampling algorithms (using Russo \& Roy's framework) can be bounded by any set of confidence bounds. For complex bandit settings, designing tight confidence bounds can be significantly harder. Whilst our theoretical results suggest that GP-TS should perform similar to GP-UCB and GP-BayesUCB, our experiments demonstrate that GP-TS obtains significantly lower regret which can likely be attributed to the point raised by Russo \& Roy. This finding is also consistent with other works for GP and non-GP bandits.
>
> **Q2 Extended Comparison in Table 1: Including the regret rates alongside the regret bounds in Table 1 would greatly enhance its utility. This addition would allow readers to quickly compare the performance of different algorithms in terms of their theoretical guarantees. An extended table with this information would provide a clearer overview of the contributions and situate the work more firmly within the existing literature.**
>
> Thank you for the suggestion, we have updated Table 1 to include the regret bounds of previous work.
>
> **Q3 Discussion of Theoretical Challenges: As mentioned in the weaknesses, a dedicated section or paragraph discussing the theoretical challenges faced in deriving the regret bounds for GP-BayesUCB, GP-UCB, and GP-TS would add significant value. This discussion could cover aspects such as handling the volatility in combinatorial settings, managing the complexities introduced by semi-bandit feedback, or other technical hurdles specific to these algorithms.**
>
> In section 3.1 and 3.2, note that we do discuss the theoretical challenges encountered although we frame it in terms of our technical contributions. Prior to introducing Lemma 3.2 respectively 3.5, we discuss the limits of the analysis from previous work and what we do to overcome it. For example in section 3.1, we describe how $\mathbb{E}[U_t(\mathbf{a}\_{t})-f(\mathbf{a}\_{t})]$ must be bound differently compared to the standard case of GP-UCB in Russo \& Roy (2014) due to the inverse error function $\text{erf}^{-1}(u)$ used by GP-BayesUCB or, in section 3.2, we point out that the volatile setting invalidates a key step in the proof of Takeno et al. (2023) and that we overcome that by analyzing the discretization error of $U_t([\mathbf{a}]_{\mathcal{D}_t}) - U_t(\mathbf{a})$.

---

### Official Review · Reviewer_mCBV · 2024-11-06

**Soundness:** 3
**Presentation:** 2
**Contribution:** 3
**Rating:** 5
**Confidence:** 3

**Summary:**

The authors derive Bayesian regret bounds for various algorithms applied to combinatorial volatile GP semi-bandit problems. Specifically, the authors derive regret bounds for 3 algorithms: GP-UCB, GP-BayesUCB, and GP-Thompson Sampling. In comparison to previous works, this is the first regret bound for GP-Bayes UCB, and in addition, extend the existing regret bounds for GP-UCB and GP-TS to infinite, volatile and combinatorial setting (which is also includes the popular contextual bandit setting).

The authors apply these algorithms to the problem of online energy-efficient navigation to demonstrate the performance of the various algorithms.

**Strengths:**

The main strengths of the paper is the theory. I do believe the GP semi-bandit problems considered in this paper are important, and having regret bounds for the algorithms discussed in this paper is also useful.

Specifically, it is nice to see sub-linear regret bound for all three algorithms.

Furthermore, I also believe that the general techniques developed here may be useful to derive regret bounds for other bandit settings.

**Weaknesses:**

1. I think the paper lacks some clarity, and the exposition can improve significantly. For example, it requires recalling previous literature to properly understand the set-up in Section 2.1: Is A a finite set? 2^A is the set of a all subsets of A? What happens when A is infinite as in Section 3.2?
2. Though the dependency on T is sub-linear, I am not sure how to view the dependency on K. Especially in the infinite case. Are there any lower bounds for these settings? It is hard to view how good or bad the bounds are with lack of comparisons.
3. Building on top of 2 above, I am curious to know if this is the best dependency on T you can get. I am used to seeing \sqrt{T} regret bounds for bandit algorithms -- is this not achievable in such settings?
4. I thought that the experimental section was too artificial. If the motivation is to solve the problem in best possible way, there are probably better ways of solving the problem (for example using RL), than naively applying the semi-bandit learning algorithms. If the point is to show the performance of various algorithms, a simple example would suffice. In my opinion, the addition of these experiments does not add any additional value to the paper, and does not change the fact that the papers main (only) contributions are the theoretical bounds.

**Questions:**

Please respond to my above concerns.

In addition, I would request the authors to add theorems / propositions after Theorems 3.2 and 3.6, without any \gamma_t and \beta_t terms. Or more generally, with as few variables as possible.

---

> ### Author Response · Authors · 2024-11-18
>
> We thank **reviewer mCBV** for their review and feedback. We are pleased to hear that the reviewer finds our theoretical regret bounds useful. Below, we address the points raised by the reviewer.
>
> **W1: I think the paper lacks some clarity, and the exposition can improve significantly. For example, it requires recalling previous literature to properly understand the set-up in Section 2.1: Is A a finite set? 2^A is the set of a all subsets of A? What happens when A is infinite as in Section 3.2?**
>
> $\mathcal{A}$ can be either a finite or infinite set, in section 3.1 we consider $|\mathcal{A}| < \infty$ and in section 3.2 we consider $|\mathcal{A}| = \infty$. (We can update the text for sec. 3.2 to state $|\mathcal{A}| = \infty$. Currently we only say it in words.)
>
> Yes, we use $2^{\mathcal{A}}$ to denote the power set. This is fairly standard notation but a clarifying sentence will be added.
>
> The combinatorial structure in our problem complicates the situation when $\mathcal{A}$ is infinite and it is worth clarifying further. The agent must select a combination of base arms but may not pick the same arm twice. To prevent issues with limit points, we therefore restrict the available base arms at round $t$ , $\mathcal{A}_t$ , to be finite. Even though $\mathcal{A}_t$ is finite, whether $\mathcal{A}$ is finite or infinite has large implications for contextual bandits. In the navigation problem, time-varying features (such as congestion, time of day or outside temperature) can only be modelled as continuous features if $\mathcal{A} = \infty$ but would be limited to discretized features if $\mathcal{A} < \infty$.
>
> **W2: Though the dependency on T is sub-linear, I am not sure how to view the dependency on K. Especially in the infinite case. Are there any lower bounds for these settings? It is hard to view how good or bad the bounds are with lack of comparisons.**:
>
> We have added additional comparisons regarding the dependency on $K$ to section 3.1.
>
> Regarding comparisons w.r.t. $K$, we do compare against the closest work by Nika et al. (2022) and obtain the same dependency. For a linear kernel, one could also compare against Wen et al. (2015) where a linear dependency on $K$ is obtained for CombLinTS and CombLinUCB although their setting may differ slightly from ours. Similar to us, their regret bound depends only on $K$ and not on the number of base arms. Therefore, we do not believe the infinite case should change anything about the dependency on $K$. For combinatorial semi-bandits with linear reward functions (but independent amrs), Merlis \& Mannor (2020) obtain a $\Omega(\sqrt{K} \log K)$ lower bound which would suggest a gap of $\sqrt{K} \log K$ for the linear kernel.
>
> Zheng Wen, Branislav Kveton, Azin Ashkan. Efficient Learning in Large-Scale Combinatorial Semi-Bandits. Proceedings of the 32nd International Conference on Machine Learning, PMLR 37:1113-1122, 2015.
>
> Nadav Merlis and Shie Mannor. Tight Lower Bounds for Combinatorial Multi-Armed Bandits. In Proceedings of Thirty Third Conference on Learning Theory, pp. 2830–2857. PMLR, 2020
>
> **W3: Building on top of 2 above, I am curious to know if this is the best dependency on T you can get. I am used to seeing $\sqrt{T}$ regret bounds for bandit algorithms -- is this not achievable in such settings?**
>
> We have added additional discussion about lower bounds w.r.t. $T$ to section 3.1.
>
> To our knowledge, there are no known lower bounds for Bayesian regret of GP-bandit algorithms in general. However, Scarlett et al. (2017) and Cai et al. (2021) discuss lower bounds for non-Bayesian regret. For the SE-kernel, they obtain $R(T) = \Omega(\sqrt{T (\log T)^{d/2}})$ which would indicate that our bounds are tight up to logarithmic factors of $T$.
>
> Note that we do compare our dependency on $T$ against the non-combinatorial and non-volatile results obtained by Takeno et al. (2023; 2024) in section 3.1 and 3.2. For the finite case, Takeno et al. obtains a $\mathcal{O}(\sqrt{T \gamma_T})$ bound which is $\mathcal{O}{\sqrt{\log T}}$ tighter than our bound. But for the infinite case our bounds match Takeno et al. For the infinite case, constructing a finite discretization is a standard technique used to prove the regret bounds in the literature but is believed to be the cause of the extra $\mathcal{O}(\sqrt{\log T})$ factor, see discussion in Takeno et al. (2023). We have also updated Table 1 to contain the regret bounds of previous work.
>
> Xu Cai, Jonathan Scarlett. On Lower Bounds for Standard and Robust Gaussian Process Bandit Optimization. Proceedings of the 38th International Conference on Machine Learning, PMLR 139:1216-1226, 2021.
>
> Jonathan Scarlett. Tight Regret Bounds for Bayesian Optimization in One Dimension. Proceedings of the 35th International Conference on Machine Learning, PMLR 80:4500-4508, 2018.

---

> ### Author Response · Authors · 2024-11-18
>
> **W4: I thought that the experimental section was too artificial. If the motivation is to solve the problem in best possible way, there are probably better ways of solving the problem (for example using RL), than naively applying the semi-bandit learning algorithms. If the point is to show the performance of various algorithms, a simple example would suffice. In my opinion, the addition of these experiments does not add any additional value to the paper, and does not change the fact that the papers main (only) contributions are the theoretical bounds.**
>
> Whilst the combinatorial semi-bandit problem can be seen as simply a special case of a reinforcement learning (RL) problem, RL formulations introduce unnecessary complexities (e.g. learning state transitions, transition-dependent rewards) that worsen the sample efficiency significantly. As an example, the actions selected by the agent will affect the traffic environment but those changes are unlikely to affect the agent itself since any edge is traversed at most once during an episode. For the problem of route and charging station selection, semi-bandit algorithms (Åkerblom et al., 2023a) have been demonstrated to scale up to graphs with orders of magnitude more nodes and edges compared to deep RL algorithms (Lee et al., 2020; Qian et al., 2019).
>
> Åkerblom et al. (2023b) introduce a semi-bandit framework that combines informed priors with efficient exploration at a low computational cost. Compared to Åkerblom et al. (2023), our formulation further improves the exploration and sample efficiency by utilizing the correlations (given by the GP) between edge weights. Doing the same for a RL problem would not be as straightforward and would lack the proven strong performance guarantees given by bandit methods.
>
> Lee et al. "Deep reinforcement learning based optimal route and charging station selection." Energies 13.23 (2020): 6255.
>
> Qian et al. "Deep reinforcement learning for EV charging navigation by coordinating smart grid and intelligent transportation system." IEEE transactions on smart grid 11.2 (2019): 1714-1723.
>
> Åkerblom et al. "A Combinatorial Semi-Bandit Approach to Charging Station Selection for Electric Vehicles." Transactions on Machine Learning Research (2023a).
>
> Åkerblom et al. "Online learning of energy consumption for navigation of electric vehicles". Artificial Intelligence, 317:103879, (2023b)
>
> **Q1: In addition, I would request the authors to add theorems / propositions after Theorems 3.2 and 3.6, without any $\gamma_t$ and $\beta_t$ terms. Or more generally, with as few variables as possible.**
>
> For the squared-exponential kernel and ignoring logarithmic terms we get $\text{BR}(T) = \mathcal{O}(K \sqrt{T})$. Note that $\gamma_T$ and $\beta_T$ are standard terms that appear in most regret bounds in the GP-bandit literature, see Srinivas et al. (2012). Additionally, in the paragraphs below Thm 3.2 and 3.6, we highlight the order of complexity that we obtain and compare the expression we get against previous works.
>
> By suggestion from reviewer 2DSA, we have updated Table 1 to include our regret bounds and those from previous works. The included regret bounds are simplified compared to those in Thms 3.2 and 3.6.

---

### Author Response · Authors · 2024-11-18
**Summary of changes made based on reviewer feedback**

- Text marked with ${\color{blue}\text{blue}}$ has been added and text marked with ${\color{red}\text{red}}$ will be removed.
- Added comparison of regret to Table 1, as suggested by reviewer 2DSA.
- Added clarifications to section 2.1 regarding the base arms $\mathcal{A}$ and the available base arms $\mathcal{A}_t$ based on feedback from reviewer mCBV.
- Added discussion of lower bound in terms of $K$ and $T$ to section 3.1 based on feedback from reviewer mCBV.
- To make space for the above changes, Figure 4 was removed from the main paper and are instead presented in Figure 5 in the appendix. Note that the results in the now removed Figure 4 was a subset of Figure 5 (previously Figure 6).

---

### Meta-Review · Area_Chair_46ef · 2024-12-23

**Metareview:**

This is a borderline paper with one reviewer being very positive and three reviewers somewhat critical. After having had a read through the reviews and discussions I believe that the criticisms are mostly about minor issues like the presentation of results. The paper presents useful results around Gaussian process semi-bandit problems and applies these techniques to a real-world problem. I believe that these contributions are valuable and worth being published.

**Additional Comments On Reviewer Discussion:**

There was some interaction between the reviewers and the authors but this interaction didn't change the opinions of the reviewers.

---

### Decision · Program_Chairs · 2025-01-22

Accept (Poster)